# Predominantly linear summation of metabotropic postsynaptic potentials follows coactivation of neurogliaform interneurons

**Attila Ozsvár, Gergely Komlósi, Gáspár Oláh, Judith Baka, Gábor Molnár, Gábor Tamás***

MTA-SZTE Research Group for Cortical Microcircuits of the Hungarian Academy of Sciences,, Department of Physiology, Anatomy and Neuroscience, University of Szeged, Szeged, Hungary

**Abstract** Summation of ionotropic receptor-mediated responses is critical in neuronal computation by shaping input-output characteristics of neurons. However, arithmetics of summation for metabotropic signals are not known. We characterized the combined ionotropic and metabotropic output of neocortical neurogliaform cells (NGFCs) using electrophysiological and anatomical methods in the rat cerebral cortex. These experiments revealed that GABA receptors are activated outside release sites and confirmed coactivation of putative NGFCs in superficial cortical layers in vivo. Triple recordings from presynaptic NGFCs converging to a postsynaptic neuron revealed sublinear summation of ionotropic $GABA_A$ responses and linear summation of metabotropic $GABA_B$ responses. Based on a model combining properties of volume transmission and distributions of all NGFC axon terminals, we predict that in 83% of cases one or two NGFCs can provide input to a point in the neuropil. We suggest that interactions of metabotropic GABAergic responses remain linear even if most superficial layer interneurons specialized to recruit $GABA_B$ receptors are simultaneously active.

***For correspondence:**
gtamas@bio.u-szeged.hu

**Competing interests:** The authors declare that no competing interests exist.

## Introduction

Each neuron in the cerebral cortex receives thousands of excitatory synaptic inputs that drive action potential (AP) output. The efficacy and timing of excitation is effectively governed by GABAergic inhibitory inputs that arrive with spatiotemporal precision onto different subcellular domains. Synchronization of GABAergic inputs appears to be crucial in structuring cellular and network excitation and behaviorally relevant rhythmic population activity (*Klausberger and Somogyi, 2008*). Diverse subpopulations of GABAergic neurons contribute to network mechanisms at different temporal windows and synchronized cells of particular interneuron types appear to fire in a stereotyped fashion (*Klausberger and Somogyi, 2008*). In general, this frequently results in coactivation of similar (and asynchronization of dissimilar) GABAergic inputs arriving to target neurons (*Jang et al., 2020*; *Karnani et al., 2016*; *Kvitsiani et al., 2013*), which leads to postsynaptic summation of GABAergic responses synchronously activated by presynaptic cells of the same type. Most GABAergic cell types exert inhibitory control through ionotropic $GABA_A$ receptors allowing $Cl^-$ ions to pass rapidly through the membrane (*Barker and Ransom, 1978*) and depending on the magnitude of GABA release and/or the number of synchronously active presynaptic interneurons, synaptic and extrasynaptic $GABA_A$ receptors could be recruited. The integration of ionotropic inhibitory signals on the surface of target cell dendrites is temporally precise and spatially specific (*Bloss et al., 2016*; *Klausberger, 2009*; *Müller et al., 2012*). Summation of ionotropic receptor-mediated responses are

extensively studied in the neocortex and predominantly characterized by nonlinear rules of interaction (*Jadi et al., 2012*; *Koch et al., 1983*; *London and Häusser, 2005*; *Qian and Sejnowski, 1990*; *Silver, 2010*). In addition to GABA$_A$ receptors, metabotropic GABA$_B$ receptor activation can occur during synchronized and/or long-lasting activation of GABAergic inputs (*Dutar and Nicoll, 1988*; *Isaacson et al., 1993*; *Mody et al., 1994*; *Scanziani, 2000*; *Thomson and Destexhe, 1999*).

Among the various interneuron subtypes identified in the neocortex neurogliaform cells (NGFCs) form a large subset of interneuron population (*Petilla Interneuron Nomenclature Group et al., 2008*; *Markram et al., 2004*; *Schuman et al., 2019*). Compared to other interneuron subtypes, NGFCs form dense axonal arborization with an unusually high presynaptic bouton density that is highly interconnected with other neighboring neurons. NGFCs are known to be especially effective in recruiting metabotropic GABA$_B$ receptors in addition to ionotropic GABA$_A$ receptors by sporadic firing using single cell triggered volume transmission in the microcircuit (*Oláh et al., 2009*; *Tamás et al., 2003*).

GABA binding to GABA$_B$ receptors catalyzes GDP/GTP exchange at the G$\alpha$ subunit and the separation of G$\beta\gamma$ (*Bettler et al., 2004*). The G$\beta\gamma$ subunits – as membrane-anchored proteins – locally diffuse in the plasma membrane and up to four G$\beta\gamma$ subunits bind cooperatively to G-protein gated inward rectifier potassium (GIRK) channels and trigger a channel opening that drives the membrane potential toward the K$^+$ reverse potential (*Dascal, 1997*; *Inanobe and Kurachi, 2014*; *Stanfield et al., 2002*; *Wang et al., 2016*; *Wickman and Clapham, 1995*). Activation of GABA$_B$ receptors by NGFCs controls the firing of dendritic spikes in the distal dendritic domain in pyramidal cells (PCs) (*Larkum et al., 1999*; *Palmer et al., 2012a*; *Wozny and Williams, 2011*) and activity in the prefrontal cortex is effectively controlled by the strong feed-forward GABA$_B$ inhibition mediated by NGFCs (*Jackson et al., 2018*). Moreover, GABA$_B$ receptors contribute to termination of persistent cortical activity (*Craig et al., 2013*) and slow inhibition contributes to theta oscillations in the hippocampus (*Capogna and Pearce, 2011*).

Relative to the summation of ionotropic responses, postsynaptic summation properties of metabotropic receptors are unexplored and to date, there has been no experimental analysis of how neurons integrate electric signals that are linked to inhibitory metabotropic receptors. We set out to test the summation of metabotropic receptor-mediated postsynaptic responses by direct measurements of convergent inputs arriving from simultaneously active NGFCs and to characterize the likelihood and arithmetics of metabotropic receptor interactions in a model of population output by incorporating experimentally determined functional and structural synaptic properties of NGFCs.

## Results

### Quantal and structural characteristics of GABAergic connections established by individual NGFCs

NGFCs are capable of activating postsynaptic receptors in the vicinity of their presynaptic boutons via volume transmission (*Oláh et al., 2009*). To gain insight into the possible effective radius of volume transmission, we characterized properties of NGFC-PC connections. In vitro simultaneous dual whole-cell patch clamp recordings were carried out on synaptically connected L1 NGFC to L2/3 PC pairs in brain slices from the somatosensory cortex of juvenile male Wistar rats. Pre- and postsynaptic cells were chosen based on their characteristic passive membrane and firing properties (*Figure 1A*) and recorded neurons were filled with biocytin, allowing post hoc anatomical reconstruction of recorded cells and estimation of putative synaptic release sites (*Figure 1B,H*). Single APs triggered in NGFCs elicit biphasic GABA$_A$ and GABA$_B$ receptor-mediated responses on the target neurons (*Tamás et al., 2003*). To determine the number of functional release sites (Nfrs), we recorded induced pluripotent stem cells (IPSCs) under different release probability by varying extracellular Ca$^{2+}$ and Mg$^{2+}$ concentrations (*Figure 1C,F*). NGFC evoked inhibitory postsynaptic potentials (IPSPs) show robust use-dependent synaptic depression, therefore we limited the intervals of AP triggered in NGFCs to 1 min (*Karayannis et al., 2010*; *Tamás et al., 2003*). We collected a dataset of n = 8 L1 NGFC to L2/3 PC pairs with an average of 65.5 ± 5.26 trials per pair and 32.75 ± 4.16 trials for a given Mg$^{2+}$/Ca$^{2+}$ concentration per conditions. The limited number of trials due to the use-dependent synaptic depression of NGFCs restricted our approach to Bayesian quantal analysis (BQA) previously shown to be robust for the estimation of quantal parameters (*Bhumbra and*

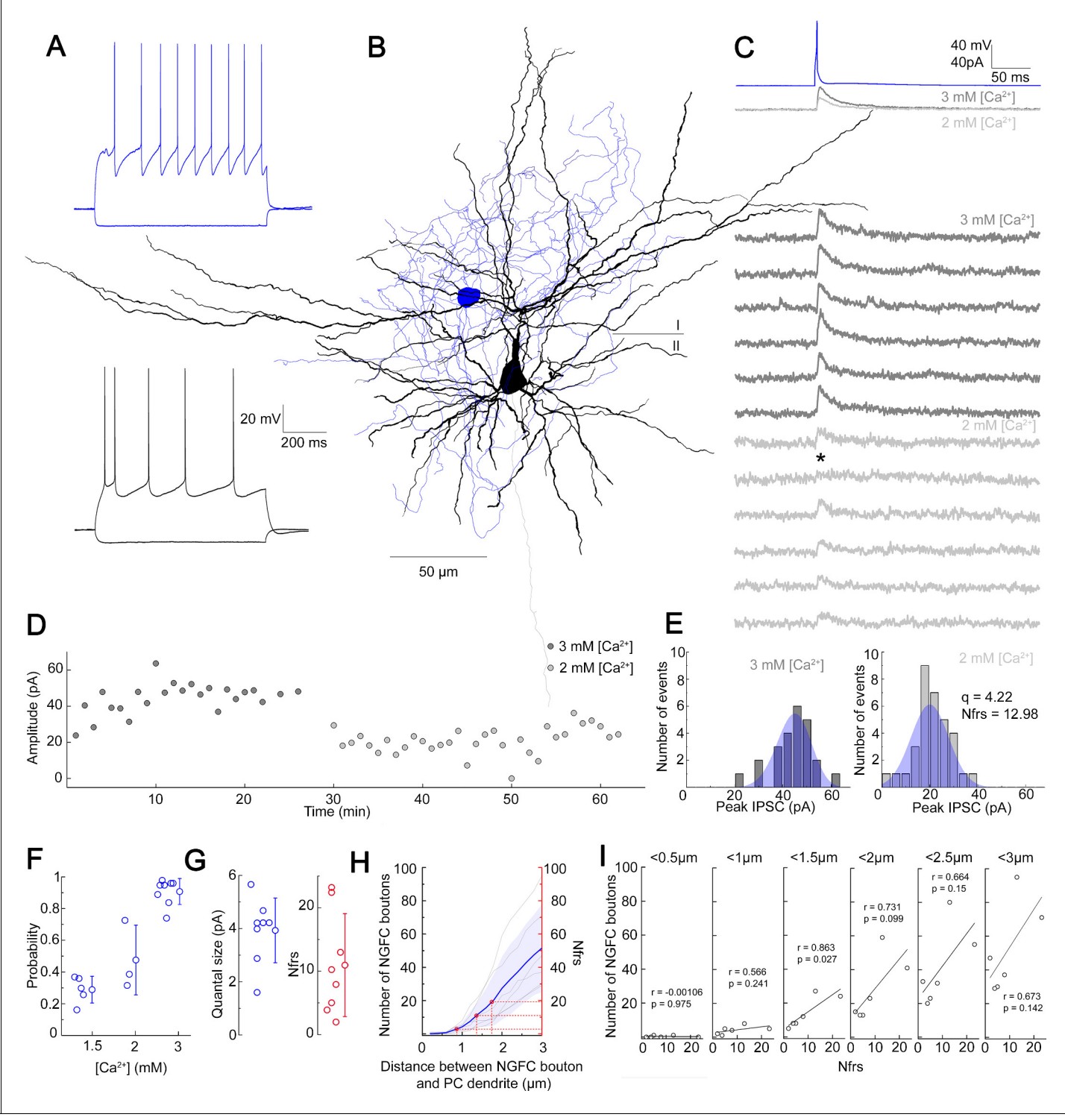

**Figure 1.** Quantal and structural characteristics of GABAergic connections established by individual neurogliaform cells (NGFCs). (A–E) Quantal and structural properties of a neurogliaform to pyramidal cell (PC) connection. (A) Firing patterns of the presynaptic L1 NGFC (blue) and postsynaptic L2/3 PC (black). (B) Three-dimensional anatomical reconstruction of a recorded L1 NGFC (soma and axon blue) and L2/3 PC (soma and dendrites black, axon gray). Horizontal line indicates the border of layer 1 (I) and layer 2 (II). (C) Presynaptic action potentials of the L1 NGFC (top, blue) elicited of unitary induced pluripotent stem cells (IPSCs) in the postsynaptic L2/3 PC at −50 mV holding potential in different $Ca^{2+}$ concentrations (middle, dark gray: 3 mM $Ca^{2+}$, 1.5 mM $Mg^{2+}$, light gray: 2 mM $Ca^{2+}$, 2 mM $Mg^{2+}$). Bottom: Representative consecutive traces of elicited unitary IPSCs. Asterisk marks synaptic transmission failure. (D) Single IPSC peak amplitudes recorded in high (3 mM $Ca^{2+}$, 1.5 mM $Mg^{2+}$, dark gray) and low release probability conditions (2 mM $Ca^{2+}$, 2 mM $Mg^{2+}$, light gray), respectively. (E) Distribution of IPSC peak amplitudes in 3 mM $Ca^{2+}$, 1.5 mM $Mg^{2+}$ (left) and 2 mM

*Figure 1 continued on next page*

*Figure 1 continued*

Ca$^{2+}$, 2 mM Mg$^{2+}$ (right), with projected binomial fits (blue). (F) Estimated release probability values in different experimental conditions (n = 8). (G) Estimated quantal size (3.93 ± 1.22 pA) and number of functional release sites (Nfrs; 10.96 ± 8.1) derived from Bayesian quantal analysis in each experiment (n = 8). (H) Number of NGFC boutons in the proximity of postsynaptic PC dendrites from anatomical reconstructions of connected NGFC to PC pairs (n = 8; gray, individual pairs; blue, average, and SD). For comparison, red lines indicate mean ± SD of Nfrs shown on panel F corresponding to distances between presynaptic NGFC boutons and PC dendrites. (I) Number of NGFC boutons counted at increasing distances from PC dendrites in NGFC to PC pairs. Correlation to Nfrs in the same pairs is best when counting boutons closer than 1.5 μm from PC dendrites.

The online version of this article includes the following figure supplement(s) for figure 1:

**Figure supplement 1.** A representative single experiment for quantal amplitude measurement during low release probability.

*Beato, 2013*). As expected, IPSC peak amplitudes were modulated by elevated (3 mM Ca$^{2+}$ and 1.5 mM Mg$^{2+}$; mean amplitude: 32.7 ± 22.17 pA; rise time: 6.47 ± 1.34 ms; decay time: 12.27 ± 2.42 ms) and reduced (1.5 mM Ca$^{2+}$ and 3 mM Mg$^{2+}$; mean amplitude: 13.58 ± 6.95 pA; rise time: 7.64 ± 3.53 ms; decay time: 11.79 ± 3.21 ms or 2 mM Ca$^{2+}$ and 2 mM Mg$^{2+}$; mean amplitude: 12.47 ± 10.9 pA; rise time: 6.95 ± 1.52 ms; decay time: 15.5 ± 10.54 ms) extracellular Ca$^{2+}$ concentrations consistent with the decline in release probability (*Figure 1D*). Distributions of IPSC amplitudes detected in paired recordings were in good agreement with the estimated quantal amplitude distribution derived from the BQA (*Figure 1E*). According to BQA, the estimated mean Nfrs was 10.96 ± 8.1 with a mean quantal size (q) of 3.93 ± 1.21 pA (*Figure 1G*). We performed n=4 experiments in which the use of low extracellular Ca$^{2+}$ reduced release probability to a level at which postsynaptic uniqantal events appeared in response to NGFC activation. We evaluated our uniquantal event detection dataset by measuring multiple parameters of each event. We measured the slope of fitted line on the initial phase of events/failures, amplitude (averaged maximum), and the area of events. Using K-means cluster analysis on the three parameters, we managed to separate the events into three groups having failures, uniquantal and multiquantal responses as separate groups. Having separated uniquantal events this way, we found that averaged quantal amplitude was 4.59±0.73 pA (n=4) which is statistically not different (p=0.68, Mann-Whitney test) from quantal amplitude measured with BQA. (*Figure 1—figure supplement 1*). Full reconstruction of functionally connected NGFC-PC pairs (n = 6) allowed comparisons of the Nfrs estimated by BQA and the number of putative release sites by counting the number of presynaptic boutons located within increasing radial distances measured from postsynaptic dendrites (*Figure 1H*). Previous experiments showed that direct synaptic junctions are not required for functional NGFC to PC connections (*Oláh et al., 2009*) and GABA reaches receptors up to 3 μm from the release site (*Farrant and Nusser, 2005*; *Overstreet-Wadiche and McBain, 2015*; *Overstreet et al., 2000*). In agreement with earlier observations (*Oláh et al., 2009*), direct appositions were not observed in most NGFC to PC pairs and the number of NGFC axonal boutons potentially involved in eliciting postsynaptic responses increased by systematically increasing the radial distance from the dendrites of PCs. Projecting the range of BQA-derived Nfrs estimates over the number of NGFC boutons putatively involved in transmission for the same connections (*Figure 1H*, red lines) suggests an effective range of 0.86–1.75 μm for nonsynaptic volume transmission from NGFCs to PCs supporting previous reports on distances covered by extrasynaptic GABAergic communication (*Farrant and Nusser, 2005*; *Overstreet-Wadiche and McBain, 2015*; *Overstreet et al., 2000*). Moreover, we detected linear correlation (r = 0.863, p = 0.027) between BQA-derived Nfrs and the number of NGFC boutons putatively involved in transmission at radial distances <1.5 μm from PC dendrites; decreasing or increasing the distance resulted in the loss of correlation (*Figure 1I*).

## Structural characteristics of GABAergic connections established by the population of layer 1 NGFCs

To have a better idea about how does the volume transmission radius potentially affect the fraction of converging outputs of L1 NGFC population to the same space, we developed a model to assess the overall output of NGFCs located in the supragranular layers of the neocortex. Unitary volume transmission by NGFCs is limited to their extremely dense axonal arborization (*Oláh et al., 2009*; *Rózsa et al., 2017*) Therefore, we determined the three-dimensional (3D) distribution of axon lengths of individual NGFCs with Sholl analysis (*Figure 2A*). By superimposing individual NGFC

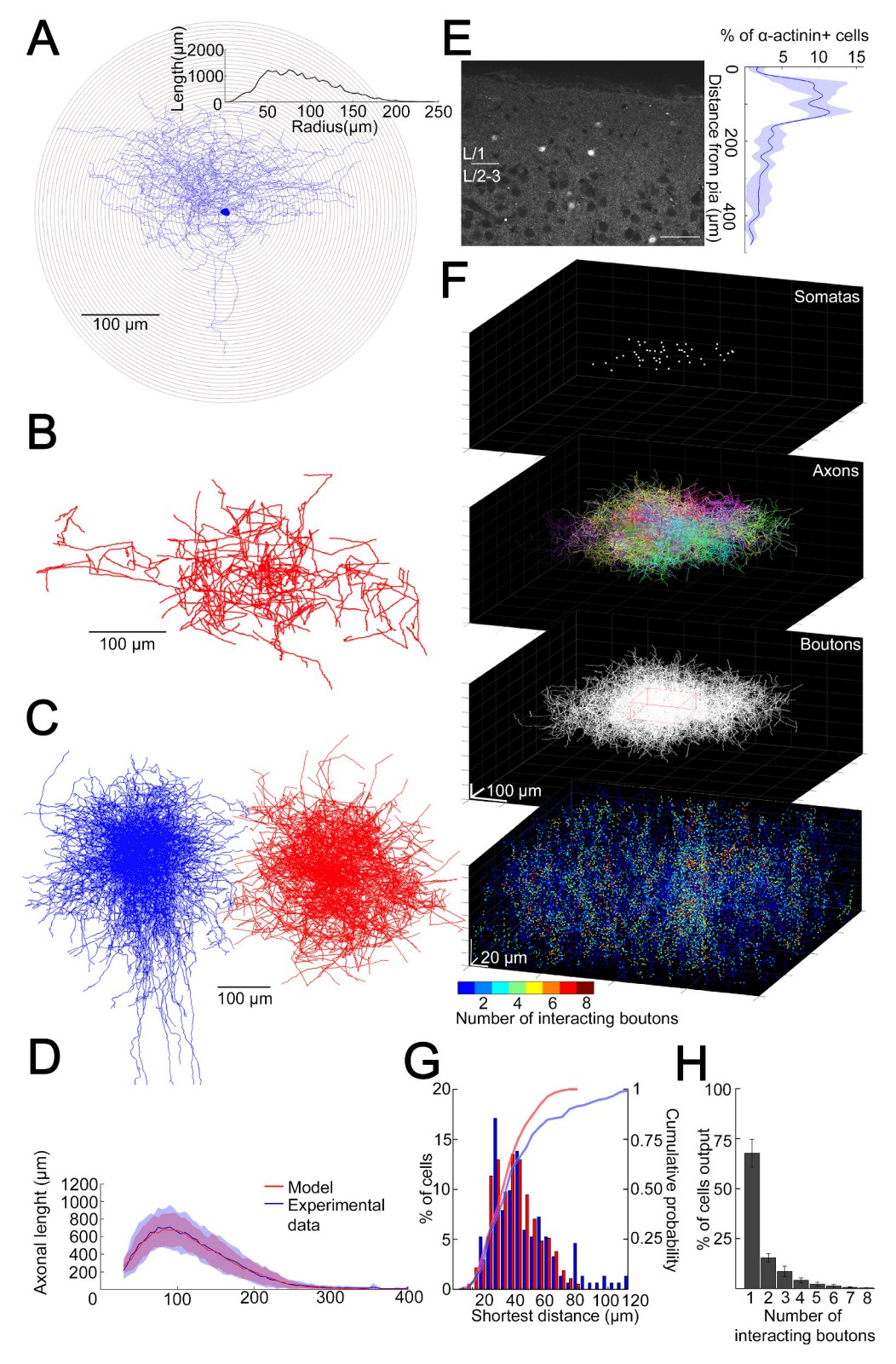

**Figure 2.** Structural characteristics of collective GABAergic output formed by the population of layer 1 neurogliaform cells (NGFCs). (**A**) Sholl analysis on the axonal arborization of an individual NGFC. Inset, axonal lengths measured in concentric shells of increasing radius (step, 10 μm). (**B**) Three-dimensional arborization of a model generated axon. (**C**) Superimposition of three-dimensionally reconstructed axonal arborizations of NGFCs (n = 16, blue) and the computer generated model NGFCs (n = 16, red) aligned at the center of somata. (**D**) Comparison of manually reconstructed axonal

*Figure 2 continued on next page*

*Figure 2 continued*

arborizations of NGFCs (n = 16; blue, mean; light blue, SD) and model generated axons (n = 52; red, mean; light red, SD). (E) Left: α-Actinin2 immunohistochemistry in supragranular layers of the neocortex (scale bar, 100 μm). Right: Distribution of α-actinin2 immunopositive somata. (F) Top: Three-dimensional model of NGFCs somata, axonal arborizations, and bouton distributions in a 354 × 354 × 140 μm³ volume. Bottom: heat map showing the number of axonal boutons interacting at distances of <1.5 μm. (G) Distribution of the shortest distance between somata in the model (red) and in α-actinin2 immunohistochemistry experiments (blue). (H) Percentage distribution of the number of interacting boutons within 1.5 μm distance from each NGFC.

The online version of this article includes the following figure supplement(s) for figure 2:

**Figure supplement 1.** Calculation of neurogliaform cells (NGFCs) interbouton interval.

reconstructions centered by their somata (n = 16; *Figure 2C*), a representative distribution of axons was calculated as a function of distance from the soma (*Figure 2D*). We also assessed the distance between axonal boutons (n = 1456) along reconstructed axons of NGFCs (n = 6) and found that interbouton distances were 3.36 ± 2.54 μm (*Figure 2—figure supplement 1*). Next, we developed an algorithm that generates model NGFCs (n = 52) by growing axon arborizations similar (p = 0.99, two-sided K-S test, *Figure 2D*) to the population of the experimentally reconstructed representative distribution of NGFC axons (n = 16) using interactions of segment lengths, branch point locations, and segment orientations while keeping the density of axonal boutons along axon segments (*Figure 2B,C*). In order to achieve a relatively complete representation of all NGFC axon terminals in a model at the population level, we performed immunhistochemical labeling of α-actinin2, which is known to label the overwhelming majority of supragranular NGFCs in the neocortex (*Uematsu et al., 2008*). Somata immunoreactive for α-actinin2 in superficial cortical layers showed distribution along the axis perpendicular to the surface of the cortex with a peak at ~50–150 μm distance from the pia mater (*Figure 2E*). According to this radial distribution and with no apparent tendency along the horizontal axis, we placed NGFC somatas in a 354 × 354 × 140 μm³ volume to create a realistic spatial model of L1 NGFC population (*Figure 2F*). 3D pairwise shortest distances between α-actinin2 + somata (n = 152) and distances between somata placed into the model space (n = 374) were similar (p = 0.51, two-sided K-S test, *Figure 2G*). We then used the axon growing algorithm detailed above from each soma position to model a population-wide distribution of NGFC axonal release sites. Quantal and structural properties of NGFC to PC connections shown above suggest a volume transmission distance of ~1.5 μm from potential sites of release (*Figure 1H,I*), thus we mapped the coverage of surrounding tissue with GABA simultaneously originating from all NGFC terminals with a 1.5 μm of transmitter diffusion in the model. Using these conditions in simulations (n = 36), less than eight NGFC axonal terminals contributed as effective GABA sources at any location in the superficial neocortex (*Figure 2H*). Moreover, these boutons originated from a limited number of presynaptic NGFCs; when considering the extreme case of population-level cooperativity, that is, when all putative NGFCs were active, most frequently a single NGFC release site serves as a GABA source (67.7 ± 7%) and potential interactions between two, three, or more different NGFCs take place in limited occasions (15.34 ± 2.1%, 8.5 ± 2.6%, and 8.45 ± 3.16%, respectively). The outcome of these simulations is consistent with earlier results suggesting that single cell-driven volume transmission covers only the close proximity of NGFCs (*Oláh et al., 2009*) but also indicates potential interactions between a restricted number of neighboring NGFCs.

## Coactivation of putative NGFCs in L1 somatosensory cortex in vivo

Transcallosal fibers establish interhemispheric inhibition that operates via GABA_B receptor activation located on apical dendrites (*Palmer et al., 2012b*) and it has been suggested that this massive GABA_B receptor recruitment in the superficial layers includes the activation of NGFCs (*Palmer et al., 2012b*). To assess the fraction of synchronously active putative NGFCs under close to physiological conditions, we applied in vivo two-photon Ca²⁺ imaging. We monitored the activity of L1 neurons bulk-loaded with calcium indicator Oregon Green BAPTA-1-AM (OGB-1-AM) (*Figure 3A*) during hindlimb stimulation, which results in the activation of transcallosal inputs in L1 of the somatosensory cortex of urethane-anesthetized rats (n = 5). Stimulation of the ipsilateral hindlimb (200 mA, 10 ms) evoked Ca²⁺ signals in a subpopulation of neurons in L1 (n = 114 neurons; n = 46 versus 68 responsive versus non-responsive neurons, respectively; data pooled from six animals; *Figure 3B and C*).

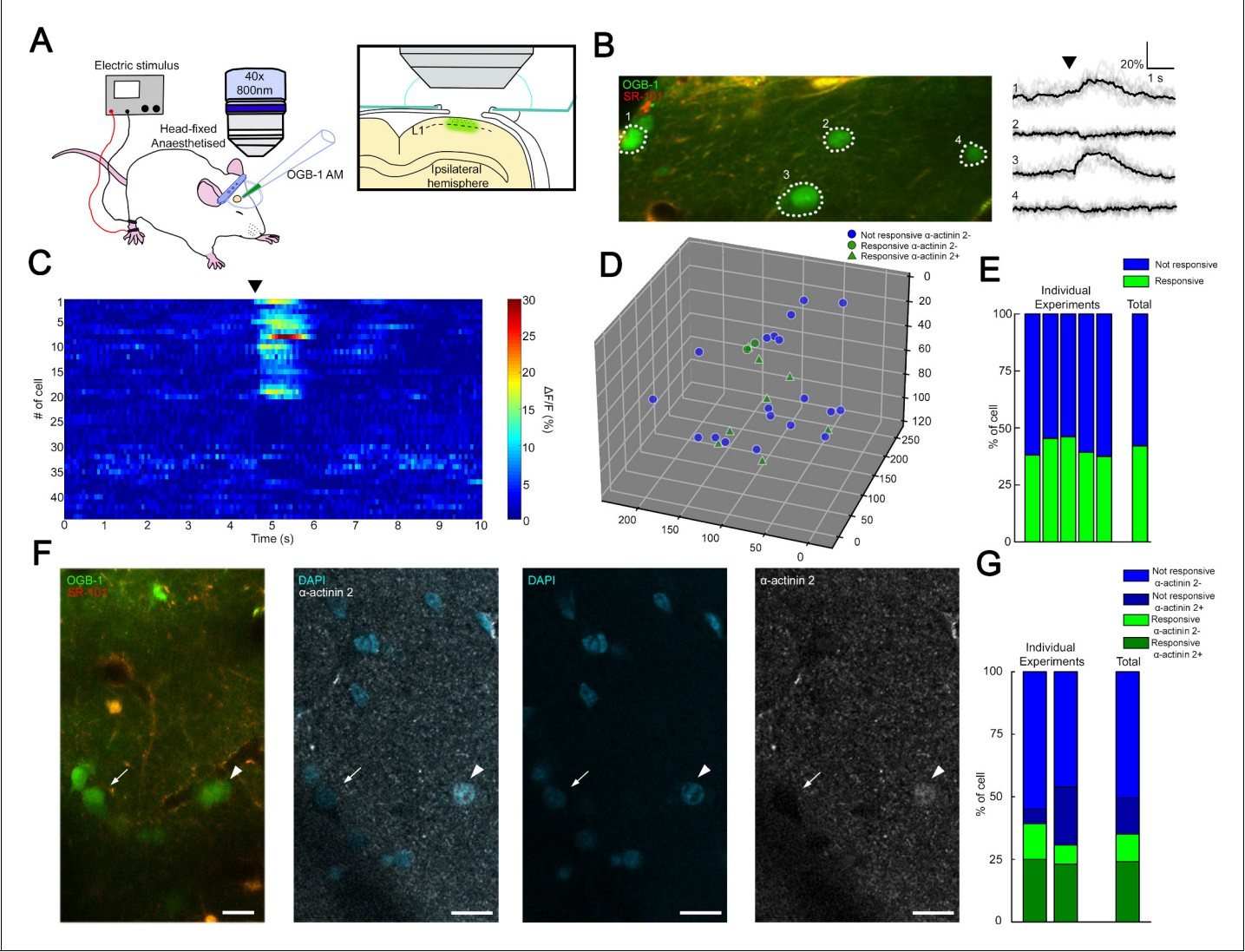

**Figure 3.** Coactivation of neurogliaform cell population in L1 somatosensory cortex in vivo. (**A**) Experimental setup. Head-fixed anesthetized rats were placed under a two-photon microscope having a cranial window above the hindlimb somatosensory cortex. OGB-1 AM and SR 101 were injected into L1. Ipsilateral hindlimb stimulation was performed with an electric stimulator. (**B**) Two-photon image of neurons that were labeled with OGB-1 in L1. SR 101 labeled astrocytes. Right: ΔF/F changes of Ca$^{2+}$ signals (gray: individual traces; black: mean of 10 consecutive traces) during series of ipsilateral stimulation (black arrowhead). Traces correspond to the marked cells. (**C**) Time-series heat map of 44 L1 interneurons evoked ΔF/F changes in Ca$^{2+}$ signals during ipsilateral hindlimb stimulation in a single experiment (black arrowhead). (**D**) Scatter plot showing the spatial location of L1 interneuron somata in a single experiment. Colors are corresponding to the responsiveness shapes to the molecular identity (not responsive and α-actinin2−, blue dots; responsive and α-actinin2−, green dots; responsive and α-actinin2+, green triangles). (**E**) Stack columns show the fraction of responsive versus not responsive cells in different experiments (n = 5 animals). Far-right columns show the mean value. (**F**) In vivo two-photon image showing imaged neurons. To the right, confocal images from the same area show immunohistochemical detection of α-actinin2+ neurons (arrowhead). α-Actinin2− cells were visualized by exclusive DAPI (4',6-diamidino-2-phenylindole) staining (arrow). Scale bar, 20 μm. (**G**) Stack columns show the proportion of α-actinin2 immunoreactivity among responsive versus not responsive cells (n = 2 animals). Far-right columns show the mean value.

The online version of this article includes the following figure supplement(s) for figure 3:

**Figure supplement 1.** Spatial relationship of responsive and not responsive L1 interneurons during ipsilateral hindlimb stimulation.

On average, 38.2 ± 5.2% of the L1 neurons were active following ipsilateral hindlimb stimulation, which is remarkably similar to the proportion found earlier (*Palmer et al., 2012b*; *Figure 3E*). To further identify L1 neurons active during hindlimb stimulation, we performed immunohistochemistry for the actin-binding protein α-actinin2 (*Uematsu et al., 2008*; *Figure 3D and F*) using the same cortical area of L1 on which two-photon imaging was performed previously. Mapping α-actinin2 positive

cells among in vivo two-photon $Ca^{2+}$ imaging monitored interneurons (*Figure 3D*) showed that closely located presumed NGFCs were active (on average 216.09 ± 77.93 μm soma to soma distance from two experiments, *Figure 3—figure supplement 1*) allowing summation of outputs by presumptive NGFCs. Cross-examination of neurons responsive/non-responsive to hindlimb stimulation versus neurons immunopositive/negative for α-actinin2 revealed that the majority of the active neurons were α-actinin2 positive (10 out of 15 neurons, 67%, n = 2 animals) and the majority of inactive neurons were α-actinin2 negative (22 out of 26 neurons, 85%, *Figure 3G*) suggesting that a substantial fraction of L1 NGFCs are activated during hindlimb stimulation. Combination of the datasets derived from the structural analysis of GABAergic connections established by the population of layer 1 NGFCs above and the in vivo mapping of coactivated putative NGFCs suggests that summation of NGFC output is feasible and, at the same time, is dominated by GABA released from boutons of a one or two NGFCs: having approximately two-thirds of NGFCs simultaneously active further suppresses the potentially synchronous contribution of three or more NGFCs (8.5% and 8.45%, see above) already constrained by structural properties in a point of the cortex.

## Summation of convergent, unitary IPSPs elicited by NGFC

Our in vivo measurements above corroborate earlier results (*Palmer et al., 2012b*) on widespread simultaneous activation of putative L1 NGFCs in response to transcallosal inputs. To directly measure the summation of converging inputs from superficial NGFCs, in vitro simultaneous triple recordings were performed from two presynaptic NGFCs and a target PC (n = 4, *Figure 4A*). First, we measured the amplitude of unitary IPSPs (n = 8) elicited by single L1 NGFCs in the target L2/3 PC and found that smaller and bigger inputs in a triplet were −1.68 ± 1.51 and −2.19 ± 1.33 mV (rise time: 4.66 ± 2 ms; half-width: 19.29 ± 5.76 ms; decay time: 8.94 ± 2.91 ms), respectively. Next we activated the two L1 NGFC inputs synchronously (0.17 ± 0.05 ms) and such coactivation resulted in moderately sublinear summation of convergent IPSPs (maximal nonlinearity, −9.1 ± 4.3%) measured as the difference of calculated (−3.81 ± 2.76 mV) and experimentally determined (−3.57 ± 2.55 mV) sums of convergent single inputs (n = 4, *Figure 4B*). These results are in line with experiments showing moderately sublinear interactions between identified, single cell evoked fast IPSPs (*Tamás et al., 2002*). Interestingly, the time course of sublinearity followed the fast, presumably $GABA_A$ receptor-mediated part of the unitary and summated IPSPs (*Figure 4C*) suggesting that ionotropic and metabotropic GABAergic components of the same input combinations might follow different rules of summation. To test the interaction of unitary $GABA_B$ receptor-mediated IPSPs directly, we repeated the experiments above with the application of the $GABA_A$ receptor antagonist gabazine (10 μM). Pharmacological experiments on the output of NGFCs are very challenging due to the extreme sensitivity of NGFC triggered IPSPs to presynaptic firing frequency (*Capogna, 2011*; *Tamás et al., 2003*) forcing us to collect the data in a different set of triple recordings (n = 8, *Figure 4D*). As expected (*Tamás et al., 2003*) unitary, gabazine-insensitive, slow IPSPs had onset latencies, rise times, and half-widths similar to $GABA_B$ receptor-mediated responses (49.42 ± 5.8, 86.95 ± 8.82, and 252.27 ± 36.92 ms, respectively, n = 16, *Figure 4E*). Peak amplitudes of converging smaller and bigger slow IPSPs were −0.66±0.22 and −0.94±0.37 mV, respectively. Synchronous activation of two presynaptic L1 NGFC converging onto the same L2/3 PC resulted in linear (−1.6 ± 6.6%) summation of slow IPSPs as peak amplitudes of calculated sums of individual inputs versus experimentally recorded compound responses were −1.58 ± 0.53 and −1.60 ± 0.55 mV, respectively (*Figure 4F*). Taken together, our triple recordings in gabazine versus control conditions suggest significantly different (p = 0.021, two-sided MW U-test) linear interactions between slow, $GABA_B$ IPSPs as opposed to sublinearly summating fast, $GABA_A$ IPSPs elicited by the same presynaptic interneuron population.

## Integration of $GABA_B$ receptor-mediated responses are not affected by HCN channel and GABA reuptake

The predominant target area of the superficial NGFCs, the distal apical dendritic membrane of PCs, express voltage-dependent hyperpolarization-activated cyclic nucleotide-gated channel 1 (HCN1) known to attenuate dendritic signals (*Berger et al., 2001*; *Kalmbach et al., 2018*; *Lörincz et al., 2002*; *Robinson and Siegelbaum, 2003*; *Sheets et al., 2011*). To investigate whether HCN1 channels contribute to mechanisms of interactions between $GABA_B$ receptor-mediated postsynaptic

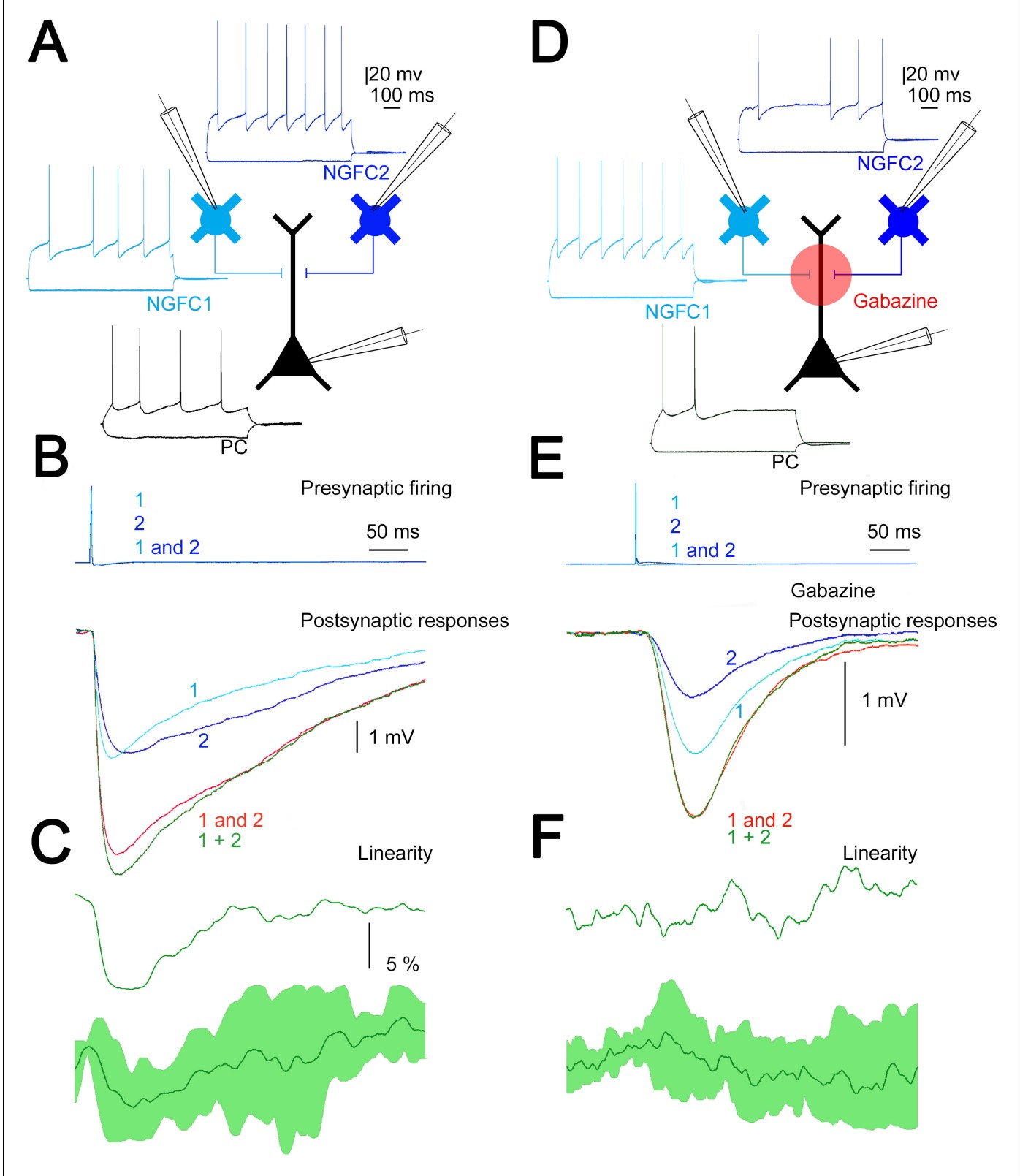

**Figure 4.** Summation of convergent, unitary inhibitory postsynaptic potentials (IPSPs) elicited by neurogliaform cells (NGFCs). (**A**) Schematic experimental setup of triplet recordings. Firing pattern of two presynaptic NGFCs (light blue and blue) and a postsynaptic pyramidal cell (PC) (black). (**B**) Action potential triggered under control conditions in the NGFCs individually (1, 2) or synchronously (1 and 2) elicited unitary (1, 2) and convergent (1 and 2) IPSPs in the postsynaptic PC. Below, the time course of the difference between the measured (1 and 2) and calculated (1 + 2) sums of

*Figure 4 continued on next page*

Figure 4 continued

convergent IPSPs. (C) The linearity of response summation in a single experiment (top) and on populations of convergent NGFC triggered IPSPs (bottom) recorded in control conditions (n = 4) (dark green, population average; light green, SD). (D) Same as experimental setup as (A) on a different set of cells but in the presence of GABA$_A$ receptor antagonist, gabazine. (E) Identical stimulation protocol as (B), note the disappearance of the difference between the measured (1 and 2) and calculated (1 + 2) sums of convergent IPSPs. (F) Same as (C), but under blocking GABA$_A$ receptors with gabazine (n = 8) (dark green, population average; light green, SD).

responses, we performed experiments on NGFC-to-PC pairs and evoked one to four APs in a single presynaptic NGFC at 100 Hz. The high stimulus frequency allows to induce GABA release with up to four APs and still remain within the time window where the presynaptic GABA$_B$ receptors are not yet activated and unable to inhibit voltage-dependent Ca$^{2+}$ channels (*Chen and van den Pol, 1998*). This ensures that short-term plasticity does not have a presynaptic effect that could hinder the release of GABA by later APs (*Karayannis et al., 2010*). This experimental configuration mimics the extreme conditions when multiple presynaptic release sites converge in a tight space and creating excessive GABA$_B$ receptor-mediated inhibition (*Figure 5A*). Triggering a single spike in the presence of gabazine (10 µM) did not saturate postsynaptic GABA$_B$ receptors since the postsynaptic response induced by two spikes was proportional to the arithmetic sum of unitary postsynaptic responses (experimental sums: −1.25 mV calculated sums: −1.26 mV; unitary IPSP rise time: 78.69 ± 27.28 ms; decay time: 56.27 ± 15.88 ms), apparently showing linear summation properties similar to triple recordings testing summation convergent inputs above. However, further increase in the number of evoked APs to 3 and 4 introduced sublinearity to summation (n = 6, 1AP: −0.63 ± 0.50 mV, 2APs: −1.25 ± 1.06 mV, 3APs: −1.53 ± 0.84 mV, 4APs: −1.61 ± 1.09 mV, *Figure 5B*; normalized values: 2APs: 2.00 ± 1.08, 3APs: 2.34 ± 1.16, 4APs: 3.17 ± 1.26, *Figure 5C*). Importantly, recordings in the presence of HCN1 channel blocker, ZD7288 (10 µM) showed summation properties similar to control, summation was linear with two APs and changed to slightly sublinear upon the 3rd to 4th spike (n = 5, 1AP: −0.82 ± 0.63 mV, 2APs: −1.59 ± 0.76 mV, 3APs: −1.66 ± 0.72 mV, 4APs: −1.90 ± 1.07 mV, unitary IPSP rise time: 97.7 ± 29.05 ms and decay time: 103.57 ± 48.94 ms; *Figure 5B*; normalized values and its comparison to control: 2APs: 2.06 ± 1.06, p = 0.983; 3APs: 1.99 ± 1.17, p = 0.362; 4APs: 2.56 ± 1.6, p = 0.336; two-sided MW U-test, *Figure 5C*). These experiments suggest that when a physiologically probable number of NGFCs are simultaneously active, HCN1 channels are locally not recruited to interfere with the summation of GABA$_B$ receptor-mediated responses.

Previous experiments suggested that a single AP in an NGFC is able to fill the surrounding extracellular space with an effective concentration of GABA (*Oláh et al., 2009*) and, in turn, extracellular GABA concentration producing GABA$_B$ receptor activation is tightly regulated via GABA transporters (GAT-1) (*Gonzalez-Burgos et al., 2009*; *Isaacson et al., 1993*; *Rózsa et al., 2017*; *Szabadics et al., 2007*). Therefore, we tested whether GAT-1 activity affects the summation of GABA$_B$ receptor-mediated responses potentially limiting the number of GABA$_B$ receptors reached by GABA released by NGFCs. Selective blockade of GAT-1 with NO-711 (10 µM) increased the amplitude of GABA$_B$ receptor-mediated IPSP; however, it did not influence summation properties (n = 6, 1AP: −1.11 ± 0.62 mV, 2APs: −2.28 ± 1.07 mV, 3APs: −3.1 ± 0.40 mV, 4APs: −3.54 ± 1.59 mV, unitary IPSP rise time: 118.85 ± 37.58 ms; decay time: 80.54 ± 40.94 ms; *Figure 5B*; normalized values and its comparison to control: 2APs: 2.06 ± 1.17, p = 0.853; 3APs: 2.36 ± 0.31, p = 0.645; 4APs: 2.97 ± 1.54, p = 0.515; two-sided MW U-test, *Figure 5C*). Accordingly, interactions between an in vivo realistic number of simultaneously active NGFCs lead to linear GABA$_B$ response summation even if increased concentration of GABA is present in the extracellular space.

## Subcellular localization of GABA$_B$ receptor-GIRK channel complex determines summation properties

High-resolution quantitative electron microscopy showed that GABA$_B$ receptors and GIRK channels are segregated on dendritic shafts; however, receptor-channel complexes colocalize on dendritic spines (*Kulik et al., 2006*). Theoretical studies suggest that the distance between the receptor and effector limits the recruitment of effector molecules to the vicinity of receptors (*Brinkerhoff et al., 2008*; *Kulik et al., 2006*), thus we asked if summation properties were influenced by the relative location of of GABA$_B$ receptors and GIRK channels when several presynaptic inputs converge. To this end, we constructed a simulation environment based on a previously published 3D

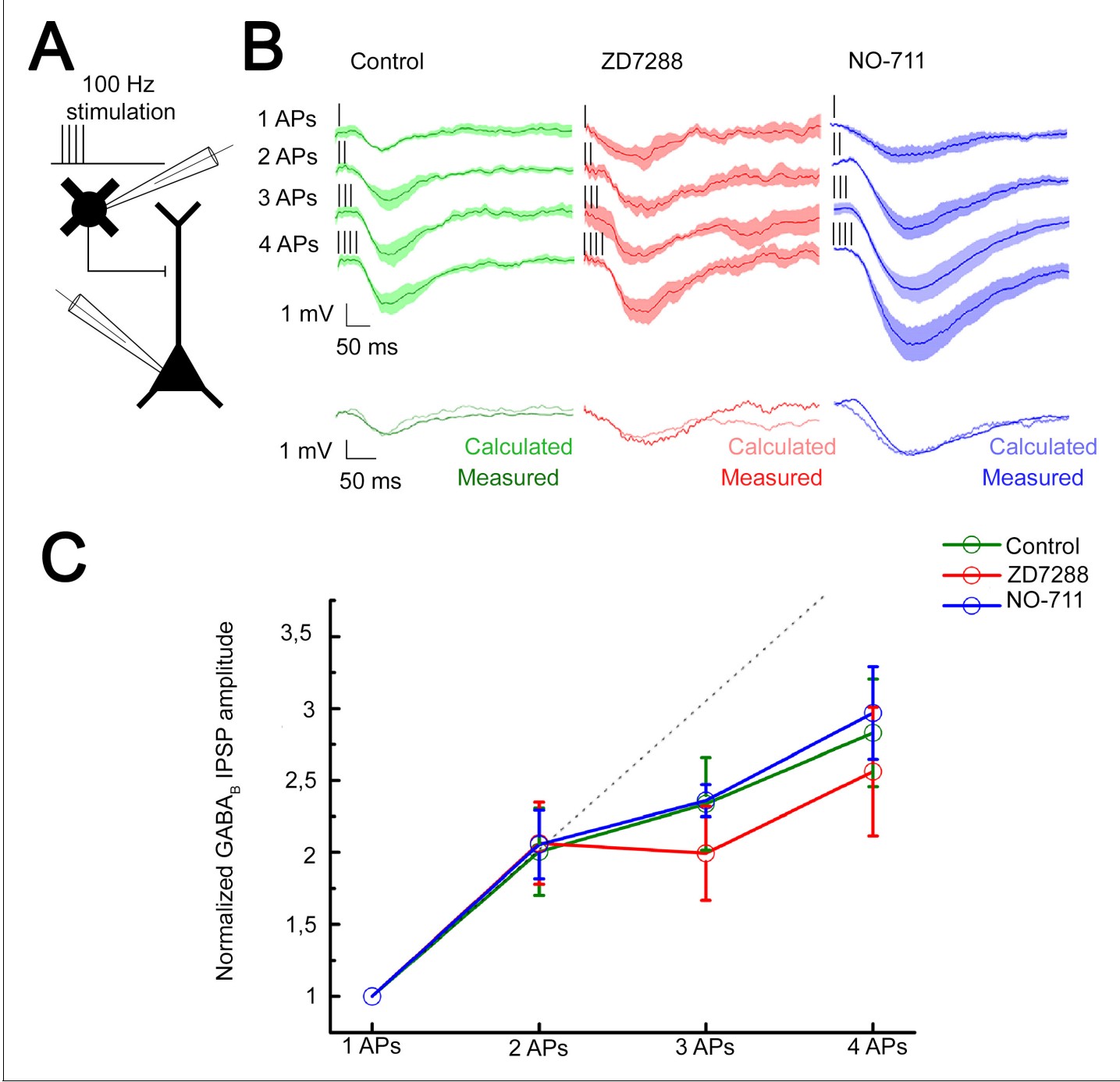

**Figure 5.** Integration of GABA$_B$ receptor-mediated responses are not affected by HCN channel and GABA reuptake. (**A**) Schematic experimental setup of paired recordings. Bursts of up to four action potentials (APs) were elicited in neurogliaform cells (NGFCs) at 100 Hz in the presence of gabazine. (**B**) NGFC to pyramidal cell (PC) paired recordings showed similar linear GABA$_B$ receptor-mediated summation under control conditions. Top: Individual traces showing inhibitory postsynaptic potential (IPSP) kinetics upon AP burst protocol (vertical lines indicating the triggered APs) during control (green traces, n = 6), in presence of hyperpolarization-activated cation (HCN) channel blocker ZD7288 (red traces, n = 5) or GABA reuptake blocker NO-711 (blue traces, n = 6). Bottom: Traces show measured IPSP from two consecutive presynaptic stimulation (measured) and the arithmetic sum of two unitary IPSP (calculated). (**C**) Summary of normalized IPSP peak amplitudes. Compare to control conditions (2APs: 2.00 ± 1.08; 3APs: 2.34 ± 1.16; 4APs: 3.17 ± 1.26), summation properties of GABA$_B$ mediated unitary IPSPs are neither affected by application of ZD7288 (2APs: 2.06 ± 1.06, p = 0.983; 3APs: 1.99 ± 1.17, p = 0.362; 4APs: 2.56 ± 1.6, p = 0.336; two-sided MW U-test) nor NO-711 (2APs: 2.06 ± 1.17, p = 0.853; 3APs: 2.36 ± 0.31, p = 0.645; 4APs: 2.9 7± 1.54, p = 0.515; two-sided MW U-test). Dashed line indicates the linearity.

reconstruction of a postsynaptic dendritic segment (*Edwards et al., 2014*) targeted by realistically positioned release sites of NGFCs (*Figure 6A and B*). Molecular interactions in this spatially realistic system were modeled using Monte Carlo algorithms to simulate movements and reactions of molecules (*Kerr et al., 2008*). Membranes of the postsynaptic dendritic segment were populated (see Materials and methods, *Figure 6—figure supplement 1*) with $GABA_B$ receptors and GIRK channels according to compartment-dependent data from SDS-digested freeze-fracture replica immunolabeling (*Kulik et al., 2006*; *Figure 6C*). Neurotransmitter diffusion in the brain is influenced by tissue tortuosity and the fraction of extracellular space in total tissue volume (*Syková and Nicholson, 2008*), thus we simulated realistic molecular diffusion in tortuous extracellular space (*Tao et al., 2005*) (see Materials and methods). The number and position of NGFC presynaptic boutons around the postsynaptic dendritic segments in the model were used according to structural characteristics of GABAergic connections established by individual NGFCs (n = 4 boutons 1.2 ± 0.7 µm from the dendrite; *Figure 1I,H*, *Figure 6—figure supplement 2*) and according to the bouton density determined for the overall output of NGFC population (*Figure 2F*). Previous work suggests that a single AP in an NGFC generates GABA concentrations of 1–60 µM lasting for 20–200 ms (*Karayannis et al., 2010*). In our model, the amount of released GABA was consistent from release event to release event and dropped off exponentially with distance from the synapse ([GABA]: 0.0 µm, 1 mM; 0.5 µm, 60 µM; 2.0 µm, 1 µM; *Figure 6—figure supplement 3*, *Supplementary file 1*, *2*). The GABA exposure time was 114.87 ± 2.1 ms with decay time constants of 11.52 ± 0.14 ms. Our modeling trials show that single AP triggered GABA release can activate a total of 5.82 ± 2.43 $GABA_B$ receptors (2.81 ± 1.55 on spine, 3.01 ± 1.71 on the shaft). Furthermore, activation of $GABA_B$ receptors triggers intracellular mechanisms and the initial GDP/GTP exchange at the Gα subunit separates the G-protein heterotrimeric protein and produces Gβγ subunits (peak number of Gβγ subunits for single AP: 338.54 ± 138.75). Lateral membrane diffusion of Gβγ subunits lead to the activation of 3.66 ± 2.17 GIRK channels in total (2.47 ± 1.88 on spine, 1.17 ± 1.26 on the shaft) in response to single AP. Next, consecutive GABA releases were induced with 10 ms delays to replicate the 100 Hz stimulation protocol used in the experiments above (*Figure 5A*). The increased GABA concentration from two sequential stimuli raised the number of active $GABA_B$ receptors to 11.29 ± 3.48 (5.57 ± 2.36 on spine, 5.72 ± 2.52 on the shaft). Three and four consecutive releases activated a total of 16.19 ± 3.88 and 20.99 ± 4.99 $GABA_B$ receptors, respectively (7.96 ± 2.74 on spine, 8.23 ± 2.97 on the shaft and 10.62 ± 3.28 on spine, 10.37 ± 3.53 on the shaft, respectively). When modeling consecutive GABA releases, massive amount of Gβγ subunits were produced together with a decline in relative production efficacy per APs, possibly due to the limited number of G-proteins serving as a substrate in the vicinity of active receptor clusters (peak number of Gβγ subunits for 2AP: 612.10 ± 171.95; 3AP: 857.78±194.14; 4AP: 1081.81 ± 229.57). Two consecutive APs resulted in the activation of 6.98 ± 3.29 GIRK channels (2.13 ± 1.63 on dendritic shaft and 4.85 ± 2.65 on spine) in the simulations. Importantly, this number of activated GIRK channels in response to two APs was close to the arithmetic sum of the number of GIRK channels activated by two single AP responses (−4.87% in total, −1.86% on spines and −9.86% on the shaft; *Figures 4E* and *5A*). Further increase in the GABA exposure proportional to three and four APs leads to the activation of 10.39 and 12.89 GIRK channels, respectively (7.01 ± 3.11 and 8.68 ± 3.46 on spines and 3.37 ± 2.08 and 4.21 ±2. 48 on the shaft, respectively). These numbers of GIRK channels corresponded to −5.68% and −13.58% of the arithmetic sum of GIRK channels activated by three and four single AP responses (−5.71% and 13.82% on spines and −4.15% and −11.16% on the shaft). The increased linearity of total GIRK channel activation relative to experimental results could be a result of several unknown properties of the cascade linking $GABA_B$ receptors to GIRK channels and/or due to vesicle depletion in terminals of NGFCs during multiple rapid release of GABA not incorporated into our model.

$GABA_B$ receptor and GIRK channel complexes located in particular subcellular compartments appeared to have different effectiveness of recruiting $GABA_B$ receptors and active GIRK channels in our simulations (*Figure 6D*). We observed different numbers of $GABA_B$ receptors activated on the shaft and spine (normalized values to 1AP: 2APs: shaft: 1.91 ± 0.84, spine: 1.98 ± 0.84, n = 534, p = 0.009; 3APs: shaft: 2.74 ± 0.99, spine: 2.84 ± 0.98, n = 1871, p = 0.173; 3APs: shaft: 3.45 ± 1.18, spine: 3.78 ± 1.17, n = 709, p< 0.005, two-sided MW U-test, *Figure 6E*). The recruitment of GIRK channels was more effective on spines compared to shafts when triggering two APs (normalized values to 1AP: shaft: 1.8±1.37, spine: 1.96±1.07, n=534, p<0.005); the trend was similar in response to

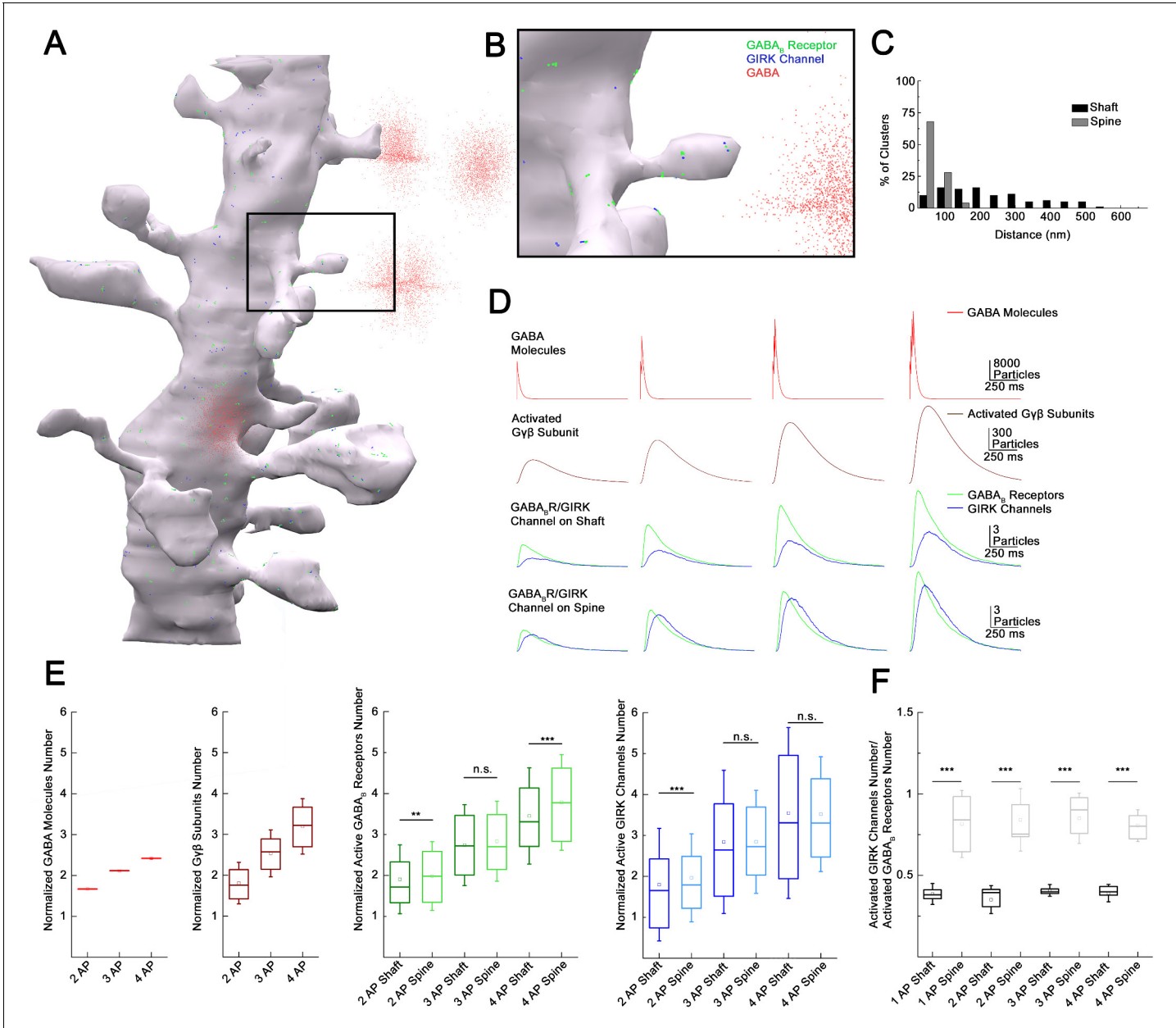

**Figure 6.** Subcellular localization of GABA_B receptor-G-protein gated inward rectifier potassium (GIRK) channel complex determines summation properties. (**A**) Visualization of the complete MCell-based model in the course of GABA release. (**B**) Magnified view of the model. (**C**) Distribution of GABA_B receptors and GIRK channel clusters on the dendritic membrane in the model (gray bars: dendritic spine; black bars: dendritic shaft). (**D**) Overview of the molecular interactions during increasing GABA release. Top to bottom: NGFC output simulated by releasing GABA (red) in the extracellular space proportional to one to four action potential (AP) stimulation. Below, the total number of produced Gβγ subunits (brown) by activated GABA_B receptors (green) located on the dendritic shaft and spine. After lateral diffusion in the plasma membrane, Gβγ subunits bind to GIRK channels (blue). (**E**) Boxplot of GABA, Gβγ subunits, GABA_B receptors, and GIRK channels quantity normalized to 1AP (GABA: 2APs: 1.67 ± 0.004, 3APs: 2.12 ± 0.005, 4APs: 2.42 ± 0.006, Gβγ subunits: 2APs: 1.85 ± 0.51, 3APs: 2.53 ± 0.57, 4APs: 3.2 ± 0.68; GABA_B receptor shaft: 2APs: 1.91 ± 0.84, 3APs: 2.74 ± 0.99, 4APs: 3.45 ± 1.18; GABA_B receptor spine: 2APs: 1.98 ± 0.84, 3APs: 2.84 ± 0.98, 4APs: 3.78 ± 1.17; GIRK channel shaft: 2APs: 1.8 ± 1.37, 3APs: 2.84 ± 1.75, 4AP: 3.55 ± 2.09; GIRK channel spine: 2APs: 1.96±1.07, 3APs: 2.84 ± 1.26, 4APs: 3.52 ± 1.4). Square indicates the mean, line shows the median inside the boxplot. (**F**) Quantification of the signaling effectiveness on the shaft and spine region of the model dendrite during increasing GABA release (1AP: shaft: 0.39 ± 0.06, spine: 0.82 ± 0.21, p < 0.005, n = 1164, two-sided MW U-test; 2AP: shaft: 0.35 ± 0.086, spine: 0.84 ± 0.19, p < 0.005, n = 534, two-sided MW U-test; 3AP: shaft: ± 0.41 ± 0.036, spine: 0.85 ± 0.15, p < 0.005, n = 1871, two-sided MW U-test; 4AP: shaft: 0.39 ± 0.05, spine: 0.81 ± 0.01, p < 0.005, n = 709, two-sided MW U-test). Square indicates the mean, line shows the median inside the boxplot.

The online version of this article includes the following figure supplement(s) for figure 6:

**Figure supplement 1.** GABA_B receptor and G-protein gated inward rectifier potassium (GIRK) channel distribution, created by a cascade reaction.

*Figure 6 continued on next page*

*Figure 6 continued*

**Figure supplement 2.** Calculation of the neurogliaform cell (NGFC) release site density for the MCell model.

**Figure supplement 3.** Estimated GABA spatial concentration profiles during multiple releases.

three and four APs, but results were not significant (normalized values to 1AP: 3AP: shaft: 2.84±1.75, spine: 2.84 ± 1.26, n = 1871, p = 0.109; 4AP: shaft: 3.55 ± 2.09, spine: 3.52 ± 1.4, n = 709, p = 0.216, two-sided MW U-test, *Figure 6E*). The compartment-specific effectiveness of signaling as the ratio of activated GIRK channels and active GABA$_B$ receptors (*Figure 6F*) shows that spines represent the preferred site of action corroborating earlier suggestions (*Qian and Sejnowski, 1990*).

## Discussion

The unique inhibitory communication via volume transmission separates NGFC interneurons from other interneuron classes in the neocortex. Numerous observations support the idea of volume transmission (*Overstreet-Wadiche and McBain, 2015*). (1) NGFC activation generates an unusually prolonged inhibition on the postsynaptic cell (*Karayannis et al., 2010*; *Mańko et al., 2012*; *Oláh et al., 2009*; *Szabadics et al., 2007*). (2) Released GABA acts on synaptic and extrasynaptic GABA receptors (*Karayannis et al., 2010*; *Oláh et al., 2009*; *Price et al., 2005*; *Tamás et al., 2003*), (3) as well as on nearby presynaptic terminals (*Oláh et al., 2009*). (4) NGFCs show a very high rate of functional coupling between the neighboring neurons (*Jiang et al., 2015*; *Oláh et al., 2009*). (5) Ultrastructural observations showed the lack of clearly defined postsynaptic elements in the apposition of the NGFC boutons (*Mańko et al., 2012*; *Oláh et al., 2009*; *Vida et al., 1998*). (6) NGFCs act on astrocytes within the reach of their axonal arborization through nonsynaptic coupling (*Rózsa et al., 2017*). The distance of effective operation through NGFC-driven volume transmission, however, is not clear. Here, we used functional and structural characterization of NGFC-PC inhibitory connections and suggest that GABA released from NGFC axonal terminals activates GABA receptors up to about ~1.8 µm, a result remarkably similar to previous estimations for the range of extrasynaptic action of synaptically released GABA (*Farrant and Nusser, 2005*; *Overstreet-Wadiche and McBain, 2015*; *Overstreet et al., 2000*). Given that our distance estimates are not based on ultrastructural evidence and we cannot exclude that fraction of anatomically defined boutons were not functional, the accuracy of our prediction for the range of volume transmission is limited. Our experiments also shed light to some quantal properties of NGFC's GABA release. These experiments are constrained by the robust use-dependent depression mediated by NGFCs (*Karayannis et al., 2010*; *Tamás et al., 2003*), therefore implementation of multiple probability fluctuation analysis (*Silver, 2003*), the gold standard for quantal analysis, was not feasible and BQA (*Bhumbra and Beato, 2013*) was needed as an alternative. The revealed linear correlation between BQA-derived Nfrs and the number of NGFC boutons putatively involved in transmission is compatible with the release of a single docked vesicle from individual NGFC boutons. However, our light microscopic methods applied in this study are insufficient for definitive claims concerning single or multivesicular release from neurogliaform boutons, especially in the light of studies highlighting the inaccuracy of comparing the Nfrs with the anatomical synaptic contacts detected by light microscopy (*Oláh et al., 2009*, *Molnár et al., 2016*, *Holler et al., 2021*).

The functional distance of volume transmission is particularly important for the characterization of interactions between NGFCs and for understanding the population output of NGFCs. Realistic representation of an entire subpopulation of neurons is considered essential for the interpretation of network functions (*Karnani et al., 2014*; *Markram et al., 2015*) and pioneering full-scale data-driven models were effective in deciphering emerging functions of interneuron populations (*Bezaire et al., 2016*). However, network diagrams addressing the function of NGFCs exclusively based on synaptic connectivity underestimate the spread of output without incorporating volume transmission by an order of magnitude (*Oláh et al., 2009*). Although the concept of blanket inhibition has been suggested for networks of interneuron populations having overlapping axonal arborizations and dense synaptic output (*Karnani et al., 2014*), our spatial model based on high-resolution reconstructions of labeled NGFCs takes the concept to its extremes and reveals an unprecedented density of release sites for a population of cortical neuron and shows that the overwhelming majority of the superficial

cortical space is effectively covered by at least one NGFC. At the same time, the redundancy of the NGFC population is limited and a single cortical spatial voxel is reached by GABA released from a limited number of individual NGFCs, ~83% of space is covered by one or two NGFCs. Our relatively simple in vivo approach to gauge potential synchronous action of NGFCs gave positive results. This is in line with earlier observations suggesting widespread action of putative NGFCs in terminating persistent activity (*Craig et al., 2013*), or powerfully suppressing dendritic Ca$^{2+}$ dynamics in L2/3 and L5 (*Palmer et al., 2012a*; *Wozny and Williams, 2011*). Strong cholinergic neuromodulation of NGFCs (*Poorthuis et al., 2018*) and frequent gap junctional coupling between NGFCs (*Simon et al., 2005*) further facilitate concerted action and are likely to play a major role in synchronizing the NGFC network (*Yao et al., 2016*).

When studying simultaneous action of NGFCs, our direct measurements of two converging NGFC inputs on L2/3 PC from simultaneous triple whole-cell patch clamp recordings revealed sublinear summation properties for ionotropic GABA$_A$ receptor-mediated responses. These results support classic theories on synaptic input interactions (*Jadi et al., 2012*; *Koch et al., 1983*; *London and Häusser, 2005*; *Qian and Sejnowski, 1990*; *Silver, 2010*) and are in line with earlier experiments measuring interactions of anatomically identified inputs converging to neighboring areas of the dendritic tree of the same postsynaptic cell (*Hao et al., 2009*; *Tamás et al., 2002*). Mechanisms of interaction between convergent inputs from NGFCs might be similar to those suggested for short-term synaptic depression of GABA$_A$ responses such as local drops in Cl$^-$ driving force and membrane conductance (*Huguenard and Alger, 1986*; *McCarren and Alger, 1985*; *Staley and Proctor, 1999*). To our knowledge, the simultaneous triple recordings of two presynaptic NGFCs targeting the same postsynaptic PC represent the first direct experimental attempt addressing the summation of metabotropic receptor-mediated postsynaptic interactions. To date, scarce computational model studies were aimed to explore the integration properties of GABA$_B$ receptor-mediated responses and suggested a highly supralinear interaction through the amplification effect of G-protein cooperativity (*Destexhe and Sejnowski, 1995*). Our experimental approach indicates linear interactions between GABA$_B$ receptor-mediated responses in case the number of converging presynaptic cells corresponds to the number of NGFCs cooperating during in vivo network operations. This suggests that converging afferents that act on inhibitory metabotropic receptors in the same postsynaptic voxel show linear or slightly sublinear summation, conserving the impact of individual inputs. However, we cannot exclude the possibility that widespread synchronization across various interneuron populations might shift the summation arithmetic in a nonlinear fashion.

Intrinsic properties of postsynaptic PCs might also contribute to the regulation of summation. HCN1 channels are known to be enriched in the distal dendrites of PCs and mediate K$^+$ cationic current activated by membrane hyperpolarization (*Kalmbach et al., 2018*; *Lörincz et al., 2002*; *Robinson and Siegelbaum, 2003*). Our experiments presented above show that summation properties in response to synchronized inputs from NGFCs are not significantly affected by HCN1 channels, presumably due to the relatively moderate local hyperpolarization arriving from NGFCs; again, further studies are needed to test the influence of additional interneuron classes coactivated together with NGFCs. We predict that further GABAergic activity is unlikely to change summation arithmetics based on our negative results when blocking the high-affinity plasma membrane GABA transporters concentrated in the perisynaptic and extrasynaptic areas (*Melone et al., 2015*) effective in modulating GABA-mediated inhibition through extrasynaptic GABA spillover (*Barbour and Häusser, 1997*; *Hamann et al., 2002*; *Scanziani, 2000*; *Szabadics et al., 2007*). Despite having similar summation arithmetics of two consecutive APs to the triple-recording configuration, it remains undefined as to what extent multiple presynaptic APs resemble synchronous activation of individual release sites. Presynaptic GABA$_B$ receptor-mediated decrease in Ca$^{2+}$ is unlikely (*Karayannis et al., 2010*); however, depletion of the readily releasable pool of vesicles leading to synaptic depression cannot be ruled out – this may contribute to the apparent inconsistency between the simulation and the experimental data concerning summation during more than three repetitive APs, since our model does not incorporate short-term synaptic depression. As suggested by pioneering simulations on the summation of GABA$_B$ receptor-mediated signaling (*Destexhe and Sejnowski, 1995*), a crucial intrinsic factor in the postsynaptic cells is the molecular cascade linking GABA$_B$ receptors to the GIRK channels through G-proteins. Our experimental evidence for close to linear or slightly sublinear summation of GABA$_B$ receptor-mediated responses suggests that even if amplification through G-proteins plays a role, it is unable to overturn local membrane or K$^+$ concentration-dependent factors promoting

sublinearity (*Dascal, 1997*; *Inanobe and Kurachi, 2014*; *Stanfield et al., 2002*; *Wickman and Clapham, 1995*). Amplification of GIRK current by G-proteins could be hampered by the need of cooperative action of up to four G-protein βγ subunits to be effective in opening GIRK channels. In addition, hyperpolarization and the accompanying relatively low $[Na^+]_i$ might also limit GIRK channel activation knowing that high $[Na^+]_i$ promotes GIRK channel opening in depolarized cells (*Wang et al., 2016*). The latter scenario might promote a brain state-dependent summation of metabotropic inhibitory signals in active neuronal networks, which remains to be tested in future experiments. On the other hand, our ultrastructural model corroborates pioneering suggestions (*Kulik et al., 2006*; *Qian and Sejnowski, 1990*) that the effect of GABA_B receptors is more prominent on dendritic spines compared to dendritic shafts, having approximately twice the number of activated GIRK channels per GABA_B receptor on spines versus shafts. Admittedly, our simulations could not cover the extensive intracellular signaling pathways known to be influenced by GABA_B receptors (*Gassmann and Bettler, 2012*; *Padgett and Slesinger, 2010*; *Terunuma, 2018*) and future availability of comprehensive transporter and extracellular space distributions of layer 1 would enrich the model (*Hrabetova et al., 2018*; *Korogod et al., 2015*; *Pallotto et al., 2015*). Nevertheless, our experiments and simulations suggest that nonsynaptic GABAergic volume transmission providing relatively homogeneous and sufficient concentrations of GABA combined with increased clustering of GABA_B receptors and on spines compared to shafts governs compartment-dependent efficacy.

Taken together, our experimental results and modeling analysis suggest that a randomly chosen location in the neuropil of layer 1 is targeted by a moderate number (usually one or two) presynaptic NGFCs. In turn, there is no apparent gap in the neurogliaform coverage of layer 1, that is, most elements of the neuropil including classic postsynaptic compartments, presynaptic terminals, or non-neuronal cells are located sufficiently close to terminals of at least one NGFC and receive GABA nonsynaptically. Interestingly, when two NGFCs which share target territory are coactivated or a single NGFC has a limited number of consecutive spikes, linear arithmetics accompany GABA_B receptor summation. This supports the hypothesis that the density and distribution of neocortical NGFCs and their axonal terminals combined with the effective range of GABAergic volume transmission appear optimized for a spatially ubiquitous and predominantly linear metabotropic GABA_B receptor summation.

## Materials and methods

### Slice preparation

Experiments were conducted to the guidelines of University of Szeged Animal Care and Use Committee (ref. no. XX/897/2018). We used young adult (19–46 days of age, (P) 23.9 ± 4.9) male Wistar rats for the electrophysiological experiments. Animals were anesthetized by inhalation of halothane, and following decapitation, 320-μm-thick coronal slices were prepared from the somatosensory cortex with a vibration blade microtome (Microm HM 650 V; Microm International GmbH, Walldorf, Germany). Slices were cut in ice-cold (4°C) cutting solution (in mM) 75 sucrose, 84 NaCl, 2.5 KCl, 1 $NaH_2PO_4$, 25 $NaHCO_3$, 0.5 $CaCl_2$, 4 $MgSO_4$, 25 D(+)-glucose, saturated with 95% $O_2$ and 5% $CO_2$. The slices were incubated in 36°C for 30 min, subsequently the solution was changed to (in mM) 130 NaCl, 3.5 KCl, 1 $NaH_2PO_4$, 24 $NaHCO_3$, 1 $CaCl_2$, 3 $MgSO_4$, 10 D(+)-glucose, saturated with 95% $O_2$ and 5% $CO_2$, and the slices were kept in it until experimental use. The solution used for recordings had the same composition except that the concentrations of $CaCl_2$ and $MgSO_4$ were 3 and 1.5 mM unless it is indicated otherwise. The micropipettes (3–5 MΩ) were filled (in mM) 126 K-gluconate, 4 KCl, 4 ATP-Mg, 0.3 GTP-Na_2, 10 HEPES, 10 phosphocreatine, and eight biocytin (pH 7.25; 300 mOsm).

### In vitro electrophysiology and pharmacology

Somatic whole-cell recordings were obtained at ~37°C from simultaneously recorded triplets and doublets of NGF and PC cell visualized by infrared differential interference contrast video microscopy at depths 60–160 μm from the surface of the slice (Zeiss Axio Examiner LSM7; Carl Zeiss AG, Oberkochen, Germany), 40× water-immersion objective (1.0 NA; Carl Zeiss AG, Oberkochen, Germany) equipped with Luigs and Neumann Junior micromanipulators (Luigs and Neumann, Ratingen,

Germany) and HEKA EPC 10 patch clamp amplifier (HEKA Elektronik GmbH, Lambrecht, Germany). Signals were filtered 5 kHz, digitalized at 15 kHz, and analyzed with Patchmaster software.

Presynaptic cells were stimulated with a brief suprathreshold current pulse (800 pA, 2–3 ms), derived in >60 s interval. In experiments, where two presynaptic NGFCs were stimulated simultaneously, the interval was increased >300 s. The stimulation sequence in which one or the other or both presynaptic NGFCs were stimulated was constantly altered, therefore the potential rundown effect or long-term potentiation would affect all three stimulation conditions equally. In the case of 100 Hz presynaptic burst stimulation, the interval was increased >300 s. During stimulation protocol, the order of triggering a set of one to four APs on the NGFC was randomized. The postsynaptic responses were normalized to the single AP in each individual set. During postsynaptic current-clamp recording, −50 mV holding current was set. The experiments were stopped if the series resistance (Rs) exceeded 35 MΩ or changed more than 20%. During postsynaptic voltage-clamp recordings, Rs and whole-cell capacitance were monitored continuously. The experiment was discarded if the compensated Rs change reached 20% during recording.

Pharmacological experiments were carried out on NGFC-PC pairs using ACSF with the following drugs: 10 µM SR 95531 hydrobromide (Tocris), 10 µM D-(-)−2-amino-5-phosphonopentanoic acid (D-AP5) (Tocris), 10 µM 2,3-dioxo-6-nitro-1,2,3,4-tetrahydrobenzo[f]quinoxaline-7-sulfonamide (NBQX) (Tocris), 10 µM 4-(N-ethyl-N-phenylamino)-1,2 dimethyl-6-(methylamino)pyrimidinium chloride (ZD7288) (Sigma-Aldrich), 10 µM 1-[2-[[(diphenylmethylene)imino]oxy]ethyl]-1,2,5,6-tetrahydro-3-pyridinecarboxylic acid hydrochloride hydrochloride (NO711) (Sigma-Aldrich).

We performed BQA by altering the extracellular $Ca^{2+}$ and $Mg^{2+}$ in two different conditions (*Bhumbra and Beato, 2013*). One of the conditions was to provide consistently a high release probability, in which the ACSF contained (in mM): 3 $Ca^{2+}$/1.5 $Mg^{2+}$. For the reduced release probability, we tested two different compositions (in mM): either 2 $Ca^{2+}$/2 $Mg^{2+}$ or 1.5 $Ca^{2+}$/ 3 $Mg^{2+}$. During BQA experiments, the ACSF solution contained the following substances: 10 µM D-AP5 (Tocris), 10 µM NBQX (Tocris). Each epoch of the BQA experiment contains a stable segment of 28 up to 42 unitary IPSCs (mean 32.75 ± 4.15). BQA experiments required at least 60 min of recording time (up to 90 min). We tested all epochs for possible long-term plasticity effect by measuring the linear correlation between IPSCs amplitude and elapsed time during the experiment, and we found no or negligible correlation (Pearson's r values from all of the experiments [n = 8] were between −0.39 and 0.46, mean −0.01 ± 0.29).

The rise time of evoked IPSCs-Ps was determined as the time interval between the points corresponding to 10% and 90% of the peak amplitude, respectively. The peak current/voltage was determined as the maximum within a window of 2 ms duration after the presynaptic AP. The decay time constant of IPSCs/Ps were measured at 67.3%. For measuring the linearity, we measured the difference of calculated and experimentally recorded IPIS peak amplitudes from converging inputs.

## Immunohistochemistry and anatomical analysis

After electrophysiological recordings, slices were fixed in a fixative containing 4% paraformaldehyde, 15% picric acid, and 1.25% glutaraldehyde in 0.1 M phosphate buffer (PB; pH = 7.4) at 4°C for at least 12 hr. After several washes in 0.1 M PB, slices were cryoprotected in 10% then 20% sucrose solution in 0.1 M PB. Slices were frozen in liquid nitrogen then thawed in PB, embedded in 10% gelatin, and further sectioned into slices of 60 µm in thickness. Sections were incubated in a solution of conjugated avidin-biotin horseradish peroxidase (ABC; 1:100; Vector Labs) in Tris-buffered saline (TBS, pH = 7.4) at 4°C overnight. The enzyme reaction was revealed by 3'3-diaminobenzidine tetrahydrochloride (0.05%) as chromogen and 0.01% $H_2O_2$ as an oxidant. Sections were post-fixed with 1% $OsO_4$ in 0.1 M PB. After several washes in distilled water, sections were stained in 1% uranyl acetate, dehydrated in ascending series of ethanol. Sections were infiltrated with epoxy resin (Durcupan [Sigma-Aldrich]) overnight and embedded on glass slices. 3D light microscopic reconstructions were carried out using Neurolucida system with a 100× objective.

## Surgery for imaging experiments

Experiments were conducted to the guidelines of University of Szeged Animal Care and Use Committee. Young adult (22–28 days of age, (P) 24.75 ± 2.75) male Wistar rats were initially anesthetized with halothane before urethane anesthesia (1.4 g/kg of body weight) was administrated

intraperitoneally. Body temperature was maintained at 37°C with a heating pad (Supertech Instruments, Pécs, Hungary). Before surgery, dexamethasone sodium phosphate (2 mg/kg of body weight) was administrated subcutaneously, and carprofen (5 mg/kg of body weight) was administrated intraperitoneally. Anesthetized animals' head were stabilized in a stereotaxic frame and headbars were attached to the skull with dental cement (Sun Medical, Mariyama, Japan). Circular craniotomy (3 mm diameter) was made above the primary somatosensory cortex, centered at 1.5 mm posterior and 2.2 mm lateral from the bregma with a high-speed dental drill (Jinme Dental, Foshan, China). Dura mater was carefully removed surgically. Finally, the craniotomy was filled with 1.5% agarose and covered with a coverslip to limit motion artifacts. The craniotomy was then submerged with HEPES buffered ACSF recording solution containing (in mM) 125 NaCl, 3.5 KCl, 10 HEPES, 1 MgSO$_4$, 1 CaCl$_2$, 0.5 D(+)-glucose, pH = 7.4.

## Two-photon calcium imaging in L1

Before covering the craniotomy with the coverslip calcium indicator, Oregon Green 488 BAPTA-1 AM (10 mM) (OGB-1 AM, Thermo Fisher Scientific) and astrocytic marker sulforhodamine 101 (1 µM) (SR101, Thermo Fisher Scientific) were pressure-injected with a glass pipette (1–2 MΩ) in L1 cortical region under the visual guide of Zeiss Axio Examiner LSM7 (Carl Zeiss AG, Oberkochen, Germany) two-photon microscope using 40× water-immersion objective (W-Plan, Carl Zeiss, Germany). Subsequently, the craniotomy was filled with agarose and covered with a coverslip. Imaging experiments were performed 1 hr after preparation. The activity of L1 interneurons was monitored during ipsilateral hindlimb electrical stimulation (Digimeter, Hertfordshire, UK, 200 mA, 10 ms). OGB-1 AM was excited at 800 nm wavelength with a femtosecond pulsing Ti:sapphire laser (Mai Tai DeepSee [Spectra-Physics, Santa Clara, CA]). In the somatosensory hindlimb region, Z-stack image series (volume size 304 × 304 × 104 µm$^3$) were acquired. Calcium signals from interneurons were obtained within this volume in full-frame mode (256 × 100 pixel), acquired at a frequency of ~20 Hz. The Ca$^{2+}$-dependent fluorescence change ∆F/F was calculated as R(t)=(F(t)−F$_0$(t))/F$_0$(t) based on *Jia et al., 2011*. The R(t) denotes the relative change of fluorescence signal, F(t) denotes the mean fluorescence of a region of interest at a certain time point, F$_0$(t) denotes the time-dependent baseline. Cells were considered responsive if there was a measurable ∆F/F change (×3 of the standard deviation of the noise) in the averaged trace of 10 trials to ipsilateral hindlimb stimulus. Image stabilization was performed by ImageJ (Fiji) software using the Image stabilizer plugin (*Li, 2008*; *Schneider et al., 2012*). At the end of the experiments, few L1 neurons were filled with biocytin containing intracellular solution to make the immunohistochemical remapping easier.

## Tissue preparation for immunohistochemistry

After imaging experiments, rats were deeply anesthetized with ketamine and xylazine. Subsequently, perfusion was performed through the aorta, first with 0.9% saline for 1 min, then with an ice-cold fixative containing 4% paraformaldehyde in 0.1 M PB (pH = 7.4) for 15 min. The whole brain was extracted and stored in 4% paraformaldehyde for 24 hr, afterward in 0.1 M PB (pH = 7.4) until slicing. Later 60-µm-thick sections were cut from the same two-photon Ca$^{2+}$ imaged brain area parallel to the pia mater and washed overnight in 0.1 M PB.

## Fluorescence immunohistochemistry and remapping

After several washes in 0.1 M PB, slices were cryoprotected with 10%, then 20% sucrose solution in 0.1M PB than frozen in liquid nitrogen. The sections were incubated for 2 hr in Alexa-488 conjugated streptavidin (1:400, Molecular Probes) solved in TBS (0.1 M; pH = 7.4) at room temperature to visualize the biocytin labeled cells. After several washes in TBS, sections were blocked in normal horse serum (NHS, 10%) made up in TBS, followed by incubation in mouse anti-α-actinin (1:20,000, Sigma-Aldrich) diluted in TBS containing 2% NHS and 0.1% Triton X-100 at room temperature for 6 hr. Following several washes in TBS, Cy3 conjugated donkey anti-mouse (1:500, Jackson ImmunoResearch) secondary antibody was used to visualize the immunoreactions. After several washes in TBS, then in 0.1 M PB, slices were counterstained with DAPI (4′,6-diamidino-2-phenylindole, Thermo Fisher Scientific). Sections were then mounted on slides in Vectashield (Vector Laboratories). Images were taken with LSM 880 confocal laser scanning microscope (Carl Zeiss AG, Oberkochen, Germany) using 40× oil-immersion objective (1.4 NA). Confocal image z-stack was tilted and panned manually to match

with the in vivo two-photon z-stack, allowing to profile imaged interneurons. During this process bio-cytin labeled neurons were used as a reference point.

## Data analysis

Electrophysiological data were analyzed with Fitmaster (HEKA Elektronik GmbH, Lambrecht, Germany), Origin 7.5 (OriginLab Corporation, Northampton, MA), IgorPro (Wavemetrics, Portland, OR). BQA experiments were analyzed using a Python written program (*Bhumbra and Beato, 2013*), incorporating NumPy and SciPy packages. Two-photon calcium imaging data were acquired with ZEN 2 (Carl Zeiss AG, Oberkochen, Germany) and analyzed with MATLAB (The MathWorks, Natick, MA), using Statistical Toolbox, Image Processing Toolbox, and custom-written scripts.

## MCell model construction

The model framework was constructed in Blender v2.7. The simulation environment contained a 3D reconstruction of a dendritic structure based on a series section of electron microscopic data (available from VolRoverN program; *Edwards et al., 2014*) and realistically positioned release sites of NGFCs. In the simulation environment, the extracellular space was also modeled by creating an array of cubic cells containing cavities based on previous work from *Tao et al., 2005*. The cubic cells have $800 \times 800$ nm$^2$ length, containing cavity that is $400 \times 400$ nm$^2$ wide, and 340 nm deep. The cubic cells and the dendritic segment were spaced 32 nm apart. The established array of cubic compartments creates an extracellular space that provides a volume fraction and tortuosity identical to the cortical brain tissue (volume fraction = 0.2 and tortuosity = 1.6). The overall dimensions of the modeled space surrounding the ultrastructurally reconstructed dendrite were $13.28 \times 13.28 \times 6.592$ μm$^3$ and the total volume was 1162.55 μm$^3$.

Simulation of GABA$_B$ receptor-GIRK channel interaction was carried out with MCell v3.4 (http://www.mcell.org) (*Kerr et al., 2008*). Custom Matlab scripts created the MDL (Model Description Language) file that required for MCell simulation. MCell simulated the release and diffusion of GABA, GABA$_B$ receptors, and GIRK channel interaction. First, to manage a biological like distribution for the receptors and channels, a reaction cascade was used (*Figure 6—figure supplement 2*). This cascade was constructed and tested in a simple simulation environment first, containing only a plane surface. At the beginning of every iteration, primary seed particles were placed on the dendritic membrane. Primary seed particles subdivided into secondary seed particles, that which then produce GABA$_B$ receptor or GIRK channel clusters. Those secondary seed particles that produce the GIRK channel clusters – which contain one to four channels – were immobile in the membrane. Meanwhile, the secondary seed particles that produce GABA$_B$ receptor clusters – which contain one to eight receptors – can diffuse laterally in the membrane. At the end, the distance was defined between the center of receptor and channel clusters by calculating the distance between each receptor and channel cluster in the 2D plane surface. Delay and the forward rate of the reaction was set to allow secondary seeds, that generate GABA$_B$ receptor clusters to diffuse to specified distance, resulting in the required GABA$_B$ receptor-GIRK channel cluster distribution as seen from *Kulik et al., 2006*. Optimization algorithm based on simulated annealing technique (*Henderson et al., 2006*; *Kirkpatrick et al., 1983*) was written in Matlab for approximating the optimal values for the delay and the forward rate of the reaction. Optimal values of delay and the forward rate of the reaction were set to allow secondary seeds, that generate GABA$_B$ receptor clusters to diffuse to specified distance, resulting in the required GABA$_B$ receptor-GIRK channel cluster distribution as seen in *Kulik et al., 2006*.

Since we were interested in the interaction between the GABA$_B$ receptors and GIRK channels, our model does not include GABA$_A$ receptors and GABA amino transporters. Previous work suggests that a single AP in the NGFC generates GABA concentration of 1–60 μM lasting for 20–200 ms (*Karayannis et al., 2010*), therefore in our model we used similar GABA concentration range of at 0.5–2 μm distance from the release sites.

Up to six MCell simulations were run with 1 μs time steps in parallel on PC with Intel (R) Core i7-4790 3.6 GHz CPU, 32 GB RAM. Total of 4278 iterations were simulated.

## Model NGFC

Based on the experimental data of NGFCs' total axonal length and its distribution, a 3D model NGFC was constructed. A custom-written Matlab algorithm generates the model NGFCs.

First single NGFC 3D models were created by expanding axonal segments from a center point that was considered to be the soma of the model cell. Three parameters were used to create model cells randomly. Total axonal length was taken from previously reconstructed cells, we used an average and SD value to determine the model NGFC's axonal length. From reconstructions, the number of branching points was the other parameter that was used as an average and SD. Finally, the orientation of the axonal segments was set, we optimized the polar angle and azimuthal angle values of the generated axonal segments in the spherical coordinate system where the origin was the soma. In each run, a Sholl analysis was done on the completed model NGFC cell, and it was compared with the experimental values of axonal distribution. The parameters were optimized to the point when statistical difference was not measurable between the model NGFCs' and reconstructed NGFCs' axonal distribution.

## Statistics

The number of experimental recordings used in each experiment is indicated in the text. Statistical tests were performed using Origin 7.5 (OriginLab Corporation, Northampton, MA) and SPSS software (IBM, Armonk, NY). Data are represented as mean ± standard deviation (SD). Data were first subject to a Shapiro-Wilk test of normality, and based on the result to the indicated parametric and non-parametric tests. Results were considered significantly different if $p < 0.05$.

## Acknowledgements

This work was supported by the ERC INTERIMPACT project, by the Eötvös Loránd Research Network, the Hungarian National Office for Research and Technology GINOP 2.3.2-15-2016-00018, Élvonal KKP 133807, the National Brain Research Program, Hungary, the National Research, Development and Innovation Office (OTKA K128863) (GT, GM); ÚNKP-20–5 – SZTE-681 New National Excellence Program of the Ministry for Innovation and Technology from the source of the National Research, Development and Innovation Fund (GM); and János Bolyai Research Scholarship of the Hungarian Academy of Sciences (GM). We would like to thank Dr Angus Silver for the useful comment on an earlier version of the manuscript, Dr Chandrajit Bajaj for the permission of using the 3D reconstruction of a dendritic structure and Éva Tóth and Nelli Ábrahám Tóth for their exceptional technical assistance.

## Additional information

### Funding

| Funder | Grant reference number | Author |
| --- | --- | --- |
| Eötvös Loránd Research Network | ELKH-SZTE Research Group for Cortical Microcircuits | Gábor Tamás |
| Hungarian Science Foundation | GINOP 2.3.2-15-2016-00018 | Gábor Tamás |
| Hungarian Science Foundation | Élvonal KKP 133807 | Gábor Tamás |
| NRDI Office | OTKA K128863 | Gábor Molnár Gábor Tamás |
| European Commission | ÚNKP-20-5 - SZTE-681 | Gábor Tamás |
| Hungarian Academy of Sciences | János Bolyai Research Scholarship | Gábor Molnár |
| Innovation and Technology Fund | New National Excellence Program | Gábor Molnár |

The funders had no role in study design, data collection and interpretation, or the decision to submit the work for publication.

## Author contributions

Attila Ozsvár, Conceptualization, Data curation, Software, Formal analysis, Validation, Investigation, Visualization, Methodology, Writing - original draft; Gergely Komlósi, Conceptualization, Data curation, Formal analysis, Validation, Investigation, Visualization, Methodology, Writing - original draft, Writing - review and editing; Gáspár Oláh, Formal analysis, Investigation, Visualization, Methodology, Writing - review and editing; Judith Baka, Formal analysis, Validation, Investigation, Visualization, Methodology, Writing - review and editing; Gábor Molnár, Formal analysis, Supervision, Investigation, Methodology, Writing - review and editing; Gábor Tamás, Conceptualization, Resources, Formal analysis, Supervision, Funding acquisition, Validation, Investigation, Visualization, Methodology, Writing - original draft, Project administration, Writing - review and editing

## Author ORCIDs

Attila Ozsvár (iD) https://orcid.org/0000-0001-5803-1174
Gábor Tamás (iD) https://orcid.org/0000-0002-7905-6001

## Ethics

Animal experimentation: Experiments were conducted to the guidelines of University of Szeged Animal Care and Use Committee (ref. no. XX/897/2018).

## Decision letter and Author response

Decision letter https://doi.org/10.7554/eLife.65634.sa1
Author response https://doi.org/10.7554/eLife.65634.sa2

# Additional files

## Supplementary files

- Supplementary file 1. Parameters used for simulation.
- Supplementary file 2. Different estimated GABA concentrations as a function of distance.
- Transparent reporting form

## Data availability

All data generated or analysed during this study are included in the manuscript and supporting files.

The following dataset was generated:

| Author(s) | Year | Dataset title | Dataset URL | Database and Identifier |
|---|---|---|---|---|
| Ozsvár A, Komlósi G, Oláh Gsr, Baka J, Molnár Gb, Tamás Gb | 2021 | Data from: Predominantly linear summation of metabotropic postsynaptic potentials follows coactivation of neurogliaform interneurons | http://dx.doi.org/10.5061/dryad.qv9s4mwf4 | Dryad Digital Repository, 10.5061/dryad.qv9s4mwf4 |

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
