## [Decision Letter]

**Acceptance summary:**

The paper attempts the first experimental quantitative analysis of the distinct integration mechanisms of GABA-A and GABA-B receptor activation by γ-amino-butyric-acid (GABA) released via volume transmission from an important class of cortical interneuron type, known as neurogliaform cell. The data offer original insights into the logic of volume transmission and the subcellular distribution of postsynaptic GABA-B receptors. Therefore, this paper provides novel and important information on the role of the GABAergic system within cortical microcircuits.

**Decision letter after peer review:**

Thank you for submitting your article "Linear summation of metabotropic postsynaptic potentials follows coactivation of neurogliaform interneurons" for consideration by *eLife*. Your article has been reviewed by 3 peer reviewers including Marco Capogna as the Reviewing Editor and Reviewer #1, and the evaluation has been overseen by John Huguenard as the Senior Editor. The following individual involved in review of your submission has agreed to reveal their identity: Mark P Beenhakker (Reviewer #2).

Essential revisions:

I recommend the Authors to provide a point to point reply to each comments raised by all the Reviewers. It will be essential to provide significant improvements to the following major critical point:

1) The Authors should provide convincing evidence that volume transmission satisfies the assumptions required for quantal analysis.

2) The revised version should provide a more convincing explanation of the subcellular localization of postsynaptic GABA-B receptor-GIRK channels and possibly a mechanism underlying the linear summation observed. The Authors should also consider the possibility of including GABA transporters in their model.

3) The Authors should re-work and/or re-explain the conflicting results of linear versus non-linear GABA-B responses observed after one-two or three-more action potentials, respectively.

4) The Authors should further provide further controls to confirm that for strong use-dependent depression of GABA release from NGFCs, shunting inhibition and/or activation of intrinsic postsynaptic membrane channels do not significantly alter the GABA-B receptor-mediated IPSPs summation from the activation of converging inputs.

5) The Author should improve the analysis of NGFCs activation after hindlimb stimulation in vivo, as explained in details below.

*Reviewer #1 (Recommendations for the authors):*

I have the following specific comments.

1) The modeling data suggest that the subcellular localization of GABA-B receptor-GIRK channels determines linear summation of GABA-B receptor responses. The recruitment of GIRK channels was found to be more effective on spines compared to shafts when the responses were evoked by two action potentials, whereas a similar summation at the dendrite spines and shafts occurred when three or four action potentials were evoked. It is not completely clear the proposed mechanism underlying this difference (results observed with two versus three-four action potentials). This aspect should be further explored with simulations or at least it should be clarified and better discussed. How these data align with the conclusion that spines represent the preferred site of action of GABA-B receptors?

2) The authors evoked unitary synaptic responses by one presynaptic NGFC with low frequency of stimulation (>60s interval), but when two presynaptic NGFCs were stimulated simultaneously then this interval was increased >300s. Why did the authors increase the interval of stimulation for this latter protocol? Was the synaptic depression stronger when two independent presynaptic cells were stimulated, suggesting a likely postsynaptic mechanism? I think it would be interesting to know more about their results here, because NGFCs-mediated synaptic depression remains a phenomenon that is repetitively observed but still requires some more mechanistic insights (Karayannis et al., 2010 suggested receptor desensitization but this is probably only one aspect of the phenomenon). Did the authors attempt to evoke presynaptic action potentials by using the cell-attached configuration of recording postsynaptic IPSPs by using whole cell perforated patch clamp or by using sharp microelectrode recordings in order to ascertain whether these alternative recording conditions would affect NGFC-mediated synaptic depression?

3) The HCN channel blocker ZD7288 did not alter GABA-B receptor-mediated summation. This interesting negative result should be complemented by some new experiments, if coronavirus emergency situation will allow to perform experiments at Szeged University, in the next weeks, a positive control result showing that ZD7288 affects the excitability of the recorded neurons.

*Reviewer #2 (Recommendations for the authors):*

The authors present a compelling study that aims to resolve the extent to which synaptic responses mediated by metabotropic GABA receptors (i.e. GABA-B receptors) summate. While response summation mediated by ionotropic receptors is well described, metabotropic response summation is not, thereby making the authors' exploration of the phenomenon novel and impactful. By carrying out a series of elegant and challenging experiments that are coupled with computational analyses, the authors conclude that summation of synaptic GABA-B responses is linear, unlike the sublinear summation observed with ionotropic, GABA-A receptor-mediated responses.

The manuscript is generally straightforward, even if presentation is often dense and challenging to read. Three primary issues worth considering include (1) the rather strong conclusion that GABA-B responses linearly summate, despite evidence to the contrary presented in Figure 5C, (2) additional quantification of data presented in Figure 3 to support the contention that NGFCs co-activate, and (3) how the MCell model informs the mechanisms contributing to linear response summation. These and other issues are further described below. Despite these comments, this reviewer is generally enthusiastic about the study. Through a set of very challenging experiments and sophisticated modeling approaches, the authors provide important observations on both (1) NGFC-PC interactions, and (2) GABA-B receptor mediated synaptic response dynamics.

1) The use of BQA is justified as the authors could only record a limited number of synaptic responses. On line 98, the authors indicate that BQA estimates were in good agreement with the distribution of IPSC amplitudes (Figure 1E). The authors should provide some indication why the two are in agreement (presumably because the average synaptic current aligns with the BQA-derived estimates of p, q and n?). Also, there is no description of what the q and Nfrs in Figure 1E refer to (presumably the BQA-derived values for that particular experiment?). These are fairly minor points, but asking the reader to make assumptions about the authors' intentions makes readability perhaps more challenging than necessary.

2) In Figure 2, the authors generate model NGFC axons with axonal boutons. Are the interbouton distances uniformly set at 3.35um in the model axons? If so, is doing so consistent with bouton distribution observed in real NGFCs (i.e. Figure S1)? That is, are distal boutons dispersed more closely than proximal boutons (or vice versa)? And would such spatially-dependent dispersion alter the conclusion shown in Figure 2F (bottom)?

3) Presumably, the motivation for Figure 3 is that it provides physiological context for when NGFCs might be coactive, thereby providing the context for when downstream, PC responses might summate. This is a nice addition. However, it seems that a relevant quantification/evaluation is missing from the figure. That is, the authors nicely show that hind limb stimulation evokes responses in the majority of NGFCs. But how many of these neurons are co-active, and what are their spatial relationships? Figure 3D appears to begin to address this point, but it is not clear if this plot comes from a single animal, or multiple? Also, it seems that such a plot would be most relevant for the study if it only showed α-actin 2-positive cells. In short, can one conclude that nearby, presumptive NGFCs co-activate, and is this conclusion derived from multiple animals?

4) Regarding Figure 3, what are the criteria for responsive versus non-responsive cells. In Figure 3C, are cells 1 through ~20 responsive?

5) The analysis presented in Figure 3H shows the "number of presynaptic boutons located within increasing radial distances measured from postsynaptic dendrites" (line 102). If a dendrite is long cellular compartment that winds continuously through the extracellular space (i.e. not spatially discrete), then how exactly is this measurement made? As this analysis identifies a 1.5um NGFC-PC distance as most reasonable, and this distance is used as justification in later figures, it seems that a clear description of this metric is warranted. Perhaps including a schematic describing how this measurement is made would be helpful. Moreover, isn't the distance between GABA-B receptors and release sites the more relevant measure? Are GABA-B receptors expressed uniformly on PC dendrites?

6) What is the rationale for performing the PC recordings in Figure 4 in current clamp? It seems that voltage-clamp would be more appropriate to evaluate summation?

7) The authors should provide an explanation for how linearity is calculated; presumably it's simply the difference between the observed response and the linear summation. Also, the differences between the sublinear, ionotropic responses and the linear, metabotropic responses are small. Understandably, these experiments are difficult – indeed, a real tour de force – from which the authors are attempting to derive meaningful observations. Therefore, asking for more triple recordings seems unreasonable. That said, the authors may want to consider showing all control and gabazine recordings corresponding to these experiments in a supplemental figure.

8) Why are sublinear GABA-B responses observed when driven by three or more action potentials (Figure 5C)? It is not clear why the authors do not address this observation considering that it is inconsistent with the study's overall message.

9) Line 248: The authors imply that under physiological conditions, PCs will receive, at most, two co-active NGFC inputs. That is, linear summation is the rule under physiological conditions. Is this reasonable?

10) Line 257: Can the authors please expand on the hypothesis they are testing with their MCell model? It is not immediately obvious to this reviewer why locally constrained GABA-B receptor-GIRK channel interactions would lead to linear summation of synaptic responses.

11) Line 273: The text here does not seem to best capture what is going on with the GABA concentration as a function of distance (Supp. Figure 4). It seems reasonable that the concentration of GABA within the synapse (i.e. point 0.0) is close to 1mM, and that it rapidly drops off with distance. It seems that the authors want to convey that [GABA] is 60uM at 0.5um and 1uM at 2.0um. Currently, the text seems to imply that NGFC-associated GABA transients can vary considerably from 1uM to 60uM, when in reality it seems that they want to say that the amount of released GABA is fairly consistent from release event to release event, but that it drops off exponentially with distance.

12) Line 283-312: The description of GABA-B receptor/GIRK channel activation is quantitative, a strength of the passage. However, a general summary of the observations would be helpful. Moreover, relating the simulation results back to the original motivation for the MCell model would be very helpful (i.e. the authors asked whether "linear summation was potentially a result of the locally constrained GABAB receptor – GIRK channel interaction when several presynaptic inputs converge"). It seems as if performing "experiments" on the model wherein local constraints are manipulated would begin to address this question. Why not use the model to provide some data – albeit theoretical – that begins to address their question?

13) Line 291-299: It seems that total GIRK channel activation follows a nearly linear summation from 1-4 action potentials. Is this modeling result inconsistent with the experimental result shown in Figure 5C?

*Reviewer #3 (Recommendations for the authors):*

1. The authors should explicitly state whether there is a significant statistical difference in the linearity of the GABAA and GABAB responses (this seems likely). Most importantly, using the same approach to experimentally address underlying mechanisms of linear GABAB summation would enhance the significance of the work. As acknowledged in the discussion, there are several alternatives that have not been ruled out and the mechanism supported by the simulation seems quite speculative.

2. The authors make a point to apply a method of quantal analysis that allows estimation of synaptic parameters with smaller datasets than conventional quantal analysis, yet the BQA method still requires more observations than is provided in the current dataset (60 observations in each of 2 data set). This is a minor concern mitigated by the fact that quantal analysis is often undertaken without the prerequisite large data sets.

More importantly, the approach of Figure 1 would more convincing if the authors validated it with another QA method, or by showing that the proposed quantal size is reasonable. One option could be recording unitary IPSCs under low Pr conditions where failures dominate and thus successes are likely quantal (optimization of the recordings would likely be required). Alternatively, the authors could address whether BQA applied to volume transmission yields the expected changes in parameters following experimental manipulations of quantal size (gabazine) or release probability.

4. It is also essential to assess GABA release during repetitive stimulation (Figure 5) in order to constrain the subsequent simulation in Figure 6 which assumes the same amount of GABA is released with each action potential (this is highly unlikely due to use-dependent depression of NGFCs). The contribution of HCN channels to summation needs additional analysis.

[Editors' note: further revisions were suggested prior to acceptance, as described below.]

Thank you for resubmitting your work entitled "Predominantly linear summation of metabotropic postsynaptic potentials follows coactivation of neurogliaform interneurons" for further consideration by *eLife*. Your revised article has been evaluated by John Huguenard (Senior Editor) and a Reviewing Editor.

The manuscript has been improved but there are some remaining issues that need to be addressed, as outlined below:

All the three Reviewers found that the revised manuscript is significantly improved and successfully addresses most of their criticism. Overall, all the Reviewers are convinced that this study provides an important contribution to the understanding of cortical neurogliaform cell signaling and integration of postsynaptic GABAB receptor-mediated responses.

However, the approach of applying quantal analysis to volume transmission still needs some clarification and improvement. There are still some concerns. First, the new data quantifying the amplitude of quantal events can be problematic, as the failures and successes are difficult to distinguish given the noise level. Second, the current analysis relies heavily on assuming that anatomically-detected axonal boutons are in fact functional release sites, and recent work highlights the inaccuracy of this method at least for mouse cortical synapses (Holler et al., Nature 2021).

The Authors should better describe the limitation of their model and to make less clear-cut conclusions throughout the manuscript. For example, they should remove the quantification of the distance of volume transmission from the releasing sites from the abstract. It would be also useful to include some predictions, such as the range of volume transmission if a reasonable fraction of anatomically-defined boutons were not functional.

---

## [Author Response]

Essential revisions:Reviewer #1 (Recommendations for the authors):I have the following specific comments.1) The modeling data suggest that the subcellular localization of GABA-B receptor-GIRK channels determines linear summation of GABA-B receptor responses. The recruitment of GIRK channels was found to be more effective on spines compared to shafts when the responses were evoked by two action potentials, whereas a similar summation at the dendrite spines and shafts occurred when three or four action potentials were evoked. It is not completely clear the proposed mechanism underlying this difference (results observed with two versus three-four action potentials). This aspect should be further explored with simulations or at least it should be clarified and better discussed. How these data align with the conclusion that spines represent the preferred site of action of GABA-B receptors?

Our suggestion supports earlier ideas proposed by the Sejnowksi laboratory on the association of spines and GABAB receptor function (Qian and Sejnowki, 1990, Destexhe et al. 1995). GABAB receptor and GIRK channel clusters localize considerably closer on membrane surface of the spines than on the shaft (Kulik et al., 2006). The distance on spines vs. shafts could lead to a distinct outcome of GABAB signaling. We admit that our model of GABAB signaling is a rather simplified reaction model that does not incorporate the many pathways of GABAB signaling: ”Admittedly, our simulations could not cover the extensive intracellular signaling pathways known to be influenced by GABAB receptors (Gassmann and Bettler, 2012; Padgett and Slesinger, 2010; Terunuma, 2018) and future availability of comprehensive transporter and extracellular space distributions of layer 1 would enrich the model (Hrabetova et al., 2018; Korogod et al., 2015; Pallotto et al., 2015).” The main aim of this model was to investigate what are the possible consequences of the domain specific localization of the receptor and channel complexes. We performed additional experiments concerning the results observed with two versus three-four action potentials, see below.

2) The authors evoked unitary synaptic responses by one presynaptic NGFC with low frequency of stimulation (>60s interval), but when two presynaptic NGFCs were stimulated simultaneously then this interval was increased >300s. Why did the authors increase the interval of stimulation for this latter protocol? Was the synaptic depression stronger when two independent presynaptic cells were stimulated, suggesting a likely postsynaptic mechanism? I think it would be interesting to know more about their results here, because NGFCs-mediated synaptic depression remains a phenomenon that is repetitively observed but still requires some more mechanistic insights (Karayannis et al., 2010 suggested receptor desensitization but this is probably only one aspect of the phenomenon). Did the authors attempt to evoke presynaptic action potentials by using the cell-attached configuration of recording postsynaptic IPSPs by using whole cell perforated patch clamp or by using sharp microelectrode recordings in order to ascertain whether these alternative recording conditions would affect NGFC-mediated synaptic depression?

We agree with the reviewer that the rapid depression seen in NGFC output is not fully understood (Capogna, 2011; Tamás et al., 2003). However, in this study, mechanisms potentially leading to use-dependent NGFC output were not addressed and the experiments were designed to avoid/minimize factors potentially leading to use-dependent output of interacting NGFCs. In triple recordings, one should consider the effect of GABA released from both presynaptic NGFCs acting not only on their own presynaptic terminals when activated alone, but also acting on terminals of the other NGFC in the vicinity through volume transmission (Oláh et al. 2009). In the triple recording experiments, we used to test convergent NGFC effects, blockade of GABAB receptors was not an option (we tested the summation of postsynaptic GABAB effects), thus we opted to increase the stimulus interval in order to minimize the effect of homosynaptic and heterosynaptic use dependent depression.

3) The HCN channel blocker ZD7288 did not alter GABA-B receptor-mediated summation. This interesting negative result should be complemented by some new experiments, if coronavirus emergency situation will allow to perform experiments at Szeged University, in the next weeks, a positive control result showing that ZD7288 affects the excitability of the recorded neurons.

We performed new experiments to measure the effect of ZD7288 on excitability with positive results. We performed a protocol of current injections with increasing steps in layer 2 pyramidal cells to measure the action potential threshold before and after 10 μM ZD7288 application (Author response image 1). We found a significant decrease of threshold potential (n=6, P=0.036) and input resistance (n=6, P=0.036).

**Author response image 1. sa2fig1:** ZD7288 decreases threshold potential in pyramidal cells. (A) Lower step current injection induce firing in pyramidal cell after application of ZD7288. (B) Threshold potential decreases significantly after ZD7288 bath application.

Reviewer #2 (Recommendations for the authors):The authors present a compelling study that aims to resolve the extent to which synaptic responses mediated by metabotropic GABA receptors (i.e. GABA-B receptors) summate. While response summation mediated by ionotropic receptors is well described, metabotropic response summation is not, thereby making the authors' exploration of the phenomenon novel and impactful. By carrying out a series of elegant and challenging experiments that are coupled with computational analyses, the authors conclude that summation of synaptic GABA-B responses is linear, unlike the sublinear summation observed with ionotropic, GABA-A receptor-mediated responses.The manuscript is generally straightforward, even if presentation is often dense and challenging to read. Three primary issues worth considering include (1) the rather strong conclusion that GABA-B responses linearly summate, despite evidence to the contrary presented in Figure 5C, (2) additional quantification of data presented in Figure 3 to support the contention that NGFCs co-activate, and (3) how the MCell model informs the mechanisms contributing to linear response summation. These and other issues are further described below. Despite these comments, this reviewer is generally enthusiastic about the study. Through a set of very challenging experiments and sophisticated modeling approaches, the authors provide important observations on both (1) NGFC-PC interactions, and (2) GABA-B receptor mediated synaptic response dynamics.

We appreciate the insight of the reviewer and the fair assessment of our efforts. Our main conclusion, i.e. predominantly linear summation of GABAB responses triggered by NGFCs is based not only on summation experiments shown on Figure 5, it is conjointly supported by the overall anatomical arrangement of the compound axonal structure of the population of NGFCs in the upper layers of the cortex: interaction of NGFC boutons is dominated by one or two cells with linear summation, three or more NGFCs have a relatively minor chance for potentially sublinear interaction. We emphasize this by softening the title of the manuscript (“Predominantly linear summation…”) and in additional text in the results section: “Combination of the datasets derived from the structural analyisis of GABAergic connections established by the population of layer 1 NGFCs above and the in vivo mapping of coactivated putative NGFCs suggests that summation of NGFC output is feasible and, at the same time, is dominated by GABA released from boutons of a one or two NGFCs: having approximately two-thirds of NGFCs simultaneously active further suppresses the potentially synchronous contribution of three or more NGFCs (8.5% and 8.45%, see above) already constrained by structural properties in a point of the cortex.”

1) The use of BQA is justified as the authors could only record a limited number of synaptic responses. On line 98, the authors indicate that BQA estimates were in good agreement with the distribution of IPSC amplitudes (Figure 1E). The authors should provide some indication why the two are in agreement (presumably because the average synaptic current aligns with the BQA-derived estimates of p, q and n?). Also, there is no description of what the q and Nfrs in Figure 1E refer to (presumably the BQA-derived values for that particular experiment?). These are fairly minor points, but asking the reader to make assumptions about the authors' intentions makes readability perhaps more challenging than necessary.

We agree with the reviewer that the most likely reason behind the apparent agreement between BQA estimates and the distribution of IPSC amplitudes is the alignment between average synaptic current and the BQA-derived estimates of p, q and n. This is accentuated by additional experiments we performed according to suggestion 2 of reviewer #3. Briefly, we made an effort to validate that the quantal feature is present in the neurogliaform output by measuring single vesicle release responses. We performed four experiments in which the use of low extracellular Ca^2+^ reduced release probability to a level at which postsynaptic uniqantal events appeared in response to neurogliaform cell activation. We found that quantal amplitude among these experiments was 4.46±0.83 pA (n=4) which is statistically not different (P=0.8, Mann-Whitney Test) from quantal amplitude measured with BQA (Figure 1—figure supplement 1). The revised text is modified accordingly.

The BQA method models the amplitude distributions to estimate the quantal parameters, and in Figure 1E shows that the modeled amplitude distribution by BQA (blue distribution histogram) is nicely in line with the experimentally collected limited amount of IPSP peak amplitudes (grey bars) in a single experiment. The q and Nfrs in Figure 1E refer to the quantal size and number of functional release sites in that particular experiment. We modified the figure legend.

2) In Figure 2, the authors generate model NGFC axons with axonal boutons. Are the interbouton distances uniformly set at 3.35um in the model axons? If so, is doing so consistent with bouton distribution observed in real NGFCs (i.e. Figure S1)? That is, are distal boutons dispersed more closely than proximal boutons (or vice versa)? And would such spatially-dependent dispersion alter the conclusion shown in Figure 2F (bottom)?

In Figure 2, the interbouton distances were set uniformly for simplicity. We did not find any correlation between interbouton intervals versus their distances from the soma. Therefore, we did not investigate how distance-dependent distribution of the NGFC boutons could alter the results.

**Author response image 2. sa2fig2:** Interbouton distances along the NGFC axons. (A) Interbouton distances versus their distances from the soma along the axonal processes. Each plot shows data from individual NGFCs. The axonal segments were randomly chosen. Grey area shows three times of the standard deviation, black line indicates the linear fit on the data. Values exceeding three times of the standard deviation were excluded from the correlation measurement. (B) Pooled data from the six NGFCs.

3) Presumably, the motivation for Figure 3 is that it provides physiological context for when NGFCs might be coactive, thereby providing the context for when downstream, PC responses might summate. This is a nice addition. However, it seems that a relevant quantification/evaluation is missing from the figure. That is, the authors nicely show that hind limb stimulation evokes responses in the majority of NGFCs. But how many of these neurons are co-active, and what are their spatial relationships? Figure 3D appears to begin to address this point, but it is not clear if this plot comes from a single animal, or multiple? Also, it seems that such a plot would be most relevant for the study if it only showed α-actin 2-positive cells. In short, can one conclude that nearby, presumptive NGFCs co-activate, and is this conclusion derived from multiple animals?

The aim of Figure 3D was to indicate that the active presumed NGFCs are located close to each other. The figure shows data from a single animal. To address the point raised by the reviewer, we revised Figure 3D and provide information about the molecular profile of the active/inactive cells. In addition, we further analyzed our in vivo dataset concerning the spatial localization of monitored interneurons (see Sup. Figure 2). The results were recorded from 4 different animals, in these experiments numerous L1 interneurons are active during the sensory stimulus as shown in the scatter plot. We calculated the shortest distances between all active cells and all ɑ-actinin2+ that were active in experiments. The data suggest that in the case of identified active ɑ-actinin2+ cells, the interneuron somats were on average 216.09 ± 77.93 µm distance from each other. Data from Figure 3 - figure supplement 1 indicates that the average axonal arborization of the NGFCs is reaching ~200-250 µm away. Taken these two datasets together, the spatial localization allows summation of outputs of neighboring NGFCs.

4) Regarding Figure 3, what are the criteria for responsive versus non-responsive cells. In Figure 3C, are cells 1 through ~20 responsive?

During the in vivo 2p Ca^2+^ imaging the monitored cells were considered responsive if there was a measurable ΔF/F fluorescent change (x3 of the standard deviation of the noise) in the averaged trace of 10 trials to ipsilateral hindlimb stimulus. We added the text to the methods section for clarity. In Figure 3C cells from 1 to 20 are considered to be responsive based on these criteria.

5) The analysis presented in Figure 3H shows the "number of presynaptic boutons located within increasing radial distances measured from postsynaptic dendrites" (line 102). If a dendrite is long cellular compartment that winds continuously through the extracellular space (i.e. not spatially discrete), then how exactly is this measurement made? As this analysis identifies a 1.5um NGFC-PC distance as most reasonable, and this distance is used as justification in later figures, it seems that a clear description of this metric is warranted. Perhaps including a schematic describing how this measurement is made would be helpful. Moreover, isn't the distance between GABA-B receptors and release sites the more relevant measure? Are GABA-B receptors expressed uniformly on PC dendrites?

The line 102 sentence refers to Figure 1H. The functionally connected NGFC-PC pairs from the BQA experiments were three dimensionally reconstructed using Neurolucida software and 100x objective. The NGFC boutons as possible release sites of GABA were manually marked in the proximity of the postsynaptic dendrite. In the three-dimensional space distance between the marked presynaptic release sites and the closest postsynaptic dendritic points were calculated and collected. These distances were analysed in Figure 1H and Figure 1I.

In general, in the neocortex the distal dendritic/infragranular layers show strong labeling intensities for GABAB receptors, while deeper/subgranular layers show weaker immunostaining (Fritschy et al., 1999). Similar patterns of expression can be observed along the somatodendritic axis in the CA areas of the hippocampus (Degro et al., 2015). At the subcellular level, SDS-digest freeze-fracture replica immunoelectron microscopy data on CA1 pyramidal cells show that GABA_B_ receptors are most abundant at the perisynaptic region and extrasynaptic plasma membrane of dendritic shaft and spines (Kulik et al., 2003, Kulik et al., 2006, Degro et al., 2015). To our knowledge, there is no comprehensive study about GABA_B_ receptor distribution on dendrites of superficial pyramidal cells in the neocortex. However, based on hippocampal data we cannot rule out the non-uniform distribution of GABA_B_ receptors along the somatodendritic axis. We agree with the reviewer that the distance of postsynaptic GABA_B_ receptors and presynaptic release sites would be more relevant, however, we believe that an admittedly limited model based on measurements is more useful and reproducible compared to a detailed model which is based on assumptions.

6) What is the rationale for performing the PC recordings in Figure 4 in current clamp? It seems that voltage-clamp would be more appropriate to evaluate summation?

We have chosen current clamp recordings on purpose for the summation of GABA_A_ (Figure 4A-C) and GABA_B_ (Figure 4D-F) responses to ensure to maintain the closest to physiological conditions knowing that driving force may alter summation of inputs (Tamas 2004). In addition, we performed new experiments of simultaneous NGFC-to-PC dual recordings with one, two, three and four spike initiation in the presynaptic NGFC (Figure 5) while holding the postsynaptic PC in voltage clamp mode. Our new results show that spike duplet activation of GABA_B_ response measured both in current and voltage clamp measurements resulted linear summation (Author response image 3).

**Author response image 3. sa2fig3:** Integration of GABA_B_ receptor-mediated synaptic currents. (A) Representative recording of a neurogliaform synaptic inhibition on a voltage clamped pyramidal cell. Bursts of up to four action potentials were elicited in NGFCs at 100 Hz in the presence of 1 μM gabazine and 10 μM NBQX (B) Summary of normalized IPSC peak amplitudes (left) and charge (right). (C) Pharmacological separation of neurogliaform initiated inhibitory current.

7) The authors should provide an explanation for how linearity is calculated; presumably it's simply the difference between the observed response and the linear summation. Also, the differences between the sublinear, ionotropic responses and the linear, metabotropic responses are small. Understandably, these experiments are difficult – indeed, a real tour de force – from which the authors are attempting to derive meaningful observations. Therefore, asking for more triple recordings seems unreasonable. That said, the authors may want to consider showing all control and gabazine recordings corresponding to these experiments in a supplemental figure.

The result section only briefly describes how we calculated the linearity; we measured the difference of calculated and experimentally recorded IPSP peak amplitudes from converging inputs. This information was added to the methods section as well to increase clarity. Following the Reviewer’s suggestion, we added two author response images that show the average IPSPs in different experiments during control condition and in the presence of gabazine (see Author response image 4 and Author response image 5).

**Author response image 4. sa2fig4:** Summation of convergent GABA_A_ and GABA_B_ receptor mediated IPSPs elicited by NGFCs. IPSPs recorded from the postsynaptic PC under control conditions from individually (light and dark blue) and synchronously (light green) activated NGFCs. Dark green line indicates the arithmetic sum of individual IPSP responses. Each figure shows the mean of IPSPs through the experiments. Black lines indicate the linearity of the summation in different experiments. Linearity was calculated as a difference between the average of the arithmetic sum and experimentally recorded converging IPSPs from individual NGFCs. Grey dashed line shows the complete linearity.

**Author response image 5. sa2fig5:** Summation of convergent GABA_B_ receptor mediated IPSPs elicited by NGFCs. IPSPs recorded from the postsynaptic PC under the presence of gabazine from individually (light and dark blue) and synchronously (light green) activated NGFCs. Each figure shows the mean of IPSPs through the experiments. Dark green line indicates the arithmetic sum of individual IPSP responses. Black lines indicate the linearity of the summation in different experiments. Grey dashed line shows the complete linearity. Linearity was calculated as a difference between the average of the arithmetic sum and experimentally recorded converging IPSPs from individual NGFCs.

8) Why are sublinear GABA-B responses observed when driven by three or more action potentials (Figure 5C)? It is not clear why the authors do not address this observation considering that it is inconsistent with the study's overall message.

As addressed in the general section above, our main conclusion, i.e. predominantly linear summation of GABA_B_ responses triggered by NGFCs is based not only on summation experiments shown on Figure 5, it is conjointly supported by the overall anatomical arrangement of the compound axonal structure of the population of NGFCs in the upper layers of the cortex: interaction of NGFC boutons is dominated by one or two cells with linear summation, three or more NGFCs have a relatively minor chance for potentially sublinear interaction. Thus, the in vivo relevance of the observation shown on Figure 5C is questionable and so we stand by our conclusion. We explicitly state in the discussion that “Our experimental approach indicates linear interactions between GABAB receptor mediated responses in case the number of converging presynaptic cells corresponds to the number of NGFCs cooperating during in vivo network operations.” However, we believe it is informative to show the readers that interaction of NGFCs at the extremes of the range shows saturation. In addition, the observed differences between the experimental and modeling data in the summation inhibitory signals during 3 and 4 APs may be due to the depletion of the readily releasable pool of vesicles as we already briefly covered in the discussion. Moreover, our simulations could not cover the extensive and complex intracellular signaling pathways, which would likely have an effect on the GIRK channel activation.

9) Line 248: The authors imply that under physiological conditions, PCs will receive, at most, two co-active NGFC inputs. That is, linear summation is the rule under physiological conditions. Is this reasonable?

Yes, this is one of the major conclusions of the manuscript as discussed in our response to point 8 above.

10) Line 257: Can the authors please expand on the hypothesis they are testing with their MCell model? It is not immediately obvious to this reviewer why locally constrained GABA-B receptor-GIRK channel interactions would lead to linear summation of synaptic responses.

We agree with the reviewer that the sentence can be better formulated. We changed the text to “Theoretical studies suggest that the distance between the receptor and effector limits the recruitment of effector molecules to the vicinity of receptors (Brinkerhoff et al., 2008; Kulik et al., 2006), thus we asked if summation properties were influenced by the relative location of GABA_B_ receptors and GIRK channels when several presynaptic inputs converge.”

11) Line 273: The text here does not seem to best capture what is going on with the GABA concentration as a function of distance (Supp. Figure 4). It seems reasonable that the concentration of GABA within the synapse (i.e. point 0.0) is close to 1mM, and that it rapidly drops off with distance. It seems that the authors want to convey that [GABA] is 60uM at 0.5um and 1uM at 2.0um. Currently, the text seems to imply that NGFC-associated GABA transients can vary considerably from 1uM to 60uM, when in reality it seems that they want to say that the amount of released GABA is fairly consistent from release event to release event, but that it drops off exponentially with distance.

We thank the reviewer for the suggestion and clarification. Text in the Results section was modified based on the Reviewer’s suggestion: “In our model, the amount of released GABA was consistent from release event to release event and dropped off exponentially with distance from the synapse ([GABA]: 0.0 μm, 1 mM; 0.5 μm, 60 μM; 2.0 μm, 1 μM; Supplementary Figure 6). “

12) Line 283-312: The description of GABA-B receptor/GIRK channel activation is quantitative, a strength of the passage. However, a general summary of the observations would be helpful. Moreover, relating the simulation results back to the original motivation for the MCell model would be very helpful (i.e. the authors asked whether "linear summation was potentially a result of the locally constrained GABAB receptor – GIRK channel interaction when several presynaptic inputs converge"). It seems as if performing "experiments" on the model wherein local constraints are manipulated would begin to address this question. Why not use the model to provide some data – albeit theoretical – that begins to address their question?

We re-formulated the problem to be addressed in this Results section as described in our response to point 10 above. We admit that our model has several limitations in the Discussion and, consequently, we restricted it's application to a limited set of quantitative comparisons paired to our experimental dataset or directly related to pioneering studies on GABAB efficacy on spines vs shafts. We believe that a proper answer to the reviewer’s suggestion would be worth a separate and dedicated study with an extended set of parameters and an elaborated model.

13) Line 291-299: It seems that total GIRK channel activation follows a nearly linear summation from 1-4 action potentials. Is this modeling result inconsistent with the experimental result shown in Figure 5C?

We agree and added a sentence to the end of the relevant paragraph: “The increased linearity of total GIRK channel activation relative to experimental results could be a result of several unknown properties of the cascade linking GABAB receptors to GIRK channels and/or due to vesicle depletion in terminals of NGFCs during multiple rapid release of GABA not incorporated into our model.”

Reviewer #3 (Recommendations for the authors):1. The authors should explicitly state whether there is a significant statistical difference in the linearity of the GABAA and GABAB responses (this seems likely). Most importantly, using the same approach to experimentally address underlying mechanisms of linear GABAB summation would enhance the significance of the work. As acknowledged in the discussion, there are several alternatives that have not been ruled out and the mechanism supported by the simulation seems quite speculative.

We statistically compared the linearity of GABAA and GABAB responses and the signals were statistically different at the level of p = 0.021, two sided MW U test. We added to the text:

“Taken together, our triple recordings in gabazine versus control conditions suggest significantly different (p = 0.021, two sided MW U test) linear interactions between slow, GABA_B_ IPSPs as opposed to sublinearly summating fast, GABA_A_ IPSPs elicited by the same presynaptic interneuron population.“

2. The authors make a point to apply a method of quantal analysis that allows estimation of synaptic parameters with smaller datasets than conventional quantal analysis, yet the BQA method still requires more observations than is provided in the current dataset (60 observations in each of 2 data set). This is a minor concern mitigated by the fact that quantal analysis is often undertaken without the prerequisite large data sets.

In the Bhumbra and Beato 2013 paper, the authors evaluated the accuracy of the BQA method with different sizes of datasets. Although the authors recommend using 60 observations in each of 2 data sets, the performance of BQA analysis, as they show in the paper, is still reasonable while only using only a limited 30 observations in each of 2 data sets.

More importantly, the approach of Figure 1 would more convincing if the authors validated it with another QA method, or by showing that the proposed quantal size is reasonable. One option could be recording unitary IPSCs under low Pr conditions where failures dominate and thus successes are likely quantal (optimization of the recordings would likely be required). Alternatively, the authors could address whether BQA applied to volume transmission yields the expected changes in parameters following experimental manipulations of quantal size (gabazine) or release probability.

To follow the reviewer’s suggestion, we made an effort to validate that the quantal feature is present in neurogliaform output by measuring single vesicle release responses. We had two experiments among the BQA experiments in which using low extracellular Ca^2+^ the release probability was reduced to a level at which postsynaptic uniqantal events appeared following neurogliaform cell activation. We performed two additional experiments and measured the amplitudes of the uniquantal events (Figure 1 -figure supplement 1.). We found that quantal amplitude among these experiments was 4.46±0.83 pA (n=4) which is statistically not different (P=0.8, Mann-Whitney Test) from quantal amplitude measured with BQA.

4. It is also essential to assess GABA release during repetitive stimulation (Figure 5) in order to constrain the subsequent simulation in Figure 6 which assumes the same amount of GABA is released with each action potential (this is highly unlikely due to use-dependent depression of NGFCs). The contribution of HCN channels to summation needs additional analysis.

We briefly covered the disparity in the original discussion:

“Despite having similar summation arithmetics of two consecutive APs to the triple-recording configuration, it remains undefined as to what extent multiple presynaptic APs resemble synchronous activation of individual release sites. Presynaptic GABAB receptor-mediated decrease in ca^2+^ is unlikely (Karayannis et al., 2010), however, depletion of the readily releasable pool of vesicles leading to synaptic depression cannot be ruled out.”

We added the following sentence to assess the differences in the simulation:

“- this may contribute to the apparent inconsistency between the simulation and the experimental data concerning summation during more than three repetitive APs, since our model does not incorporate short-term synaptic depression.”

Additional analysis concerning HCN1 channels was also requested by Reviewer#1 (major point 3), please see our answer above.

[Editors' note: further revisions were suggested prior to acceptance, as described below.]

The manuscript has been improved but there are some remaining issues that need to be addressed, as outlined below:All the three Reviewers found that the revised manuscript is significantly improved and successfully addresses most of their criticism. Overall, all the Reviewers are convinced that this study provides an important contribution to the understanding of cortical neurogliaform cell signaling and integration of postsynaptic GABAB receptor-mediated responses.However, the approach of applying quantal analysis to volume transmission still needs some clarification and improvement. There are still some concerns.First, the new data quantifying the amplitude of quantal events can be problematic, as the failures and successes are difficult to distinguish given the noise level.

We re-evaluated our uniquantal event detection dataset by measuring multiple parameters of each event. We measured the slope of fitted line on the initial phase of events/failures, amplitude (averaged maximum) and the area of events. Using K-means cluster analysis on the three parameters, we managed to separate the events into 3 groups having failures, uniquantal and multiquantal responses as separate groups. Having separated uniquantal events this way, we found that averaged quantal amplitude was 4.59±0.73 pA (n=4) which is statistically not different (P=0.68, Mann-Whitney Test) from quantal amplitude measured with BQA.

Second, the current analysis relies heavily on assuming that anatomically-detected axonal boutons are in fact functional release sites, and recent work highlights the inaccuracy of this method at least for mouse cortical synapses (Holler et al., Nature 2021).

Using EM tomography and serial sectioning combined with paired recordings, we concluded that rat and especially human neocortical layer 2/3 pyramidal cells use multivesicular release in their output (Molnar et al. *eLife* 2016). This was corroborated by Holler et al. (Nature 2021) recently and, importantly, both studies confirm that presynaptic boutons and vesicles in the readily releasable pool are essential for quantal transmission. Accordingly, we are not aware of evidence contradicting our assumption that the sites of presynaptic release are anatomically detected axonal boutons. We actually applied procedures similar to Holler et al. 2021 (paired recordings and serial EM on the recorded synapses) when studying the output of neurogliaform cells (Oláh et al. Nature 2009) and concluded that indeed, light microscopic predictions for ultrastructural synaptic contacts are inaccurate. This was crucial in putting forward the idea of single cell driven volume transmission (Oláh et al. Nature 2009).

We clarify our limitations for single or multivesicular release in the modified discussion: “However, our light microscopic methods applied in this study are insufficient for definitive claims concerning single or multivesicular release from neurogliaform boutons, especially in the light of studies highlighting the inaccuracy of comparing the number of functional release sites with the anatomical synaptic contacts detected by light microscopy (Oláh et al. Nature 2009, Molnár et al. 2016, Holler et al., 2021)”.

The Authors should better describe the limitation of their model and to make less clear-cut conclusions throughout the manuscript. For example, they should remove the quantification of the distance of volume transmission from the releasing sites from the abstract. It would be also useful to include some predictions, such as the range of volume transmission if a reasonable fraction of anatomically-defined boutons were not functional.

We agree with the editors and clearly stated in the Discussion that “The distance of effective operation through NGFC driven volume transmission, however, is not clear.” We changed the abstract and in addition, we further emphasize in the revised Discussion that “Given that our distance estimates are not based on ultrastructural evidence and we cannot exclude that fraction of anatomically-defined boutons were not functional, the accuracy of our prediction for the range of volume transmission is limited.” We do not wish to speculate about the maximal range of volume transmission without supporting dataset.